# Simulation of climate, ice sheets and $CO_2$ evolution during the last four glacial cycles with an Earth system model of intermediate complexity

Andrey Ganopolski[1] and Victor Brovkin[2]

[1]Potsdam Institute for Climate Impact Research (PIK), Potsdam, Germany. ganopolski@pik-potsdam.de

[2]Max Plank Institute for Meteorology, Hamburg, Germany; also a guest scientist at PIK

*Correspondence to*: Andrey Ganopolski (ganopolski@pik-potsdam.de)

In spite of significant progress in paleoclimate reconstructions and modeling of different aspects of the past glacial cycles, the mechanisms which transform regional and seasonal variations in solar insolation into long-term and global-scale glacial-interglacial cycles are still not fully understood, in particular, for $CO_2$ variability. Here using the Earth system model of intermediate complexity CLIMBER-2 we performed simulations of co-evolution of climate, ice sheets and carbon cycle over the last 400,000 years using the orbital forcing as the only external forcing. The model simulates temporal dynamics of $CO_2$, global ice volume and other climate system characteristics in good agreement with paleoclimate reconstructions. These results provide strong support to the idea that long and strongly asymmetric glacial cycles of the late Quaternary represent a direct but strongly nonlinear response of the Northern Hemisphere ice sheets to the orbital forcing. This direct response is strongly amplified and globalized by the carbon cycle feedbacks. Using simulations performed with the model in different configurations, we also analyze the role of individual processes and sensitivity to the choice of model parameters. While many features of simulated glacial cycles are rather robust, some details of $CO_2$ evolution, especially during glacial terminations, are sensitive to the choice of model parameters. Specifically, we found two major regimes of $CO_2$ changes during terminations: in the first one, when the recovery of the Atlantic meridional overturning circulation (AMOC) occurs only at the end of the termination, a pronounced overshoot in $CO_2$ concentration occurs at the beginning of the interglacial and $CO_2$ remains almost constant during interglacial or even decline towards the end, resembling Eemian $CO_2$ dynamics. However, if the recovery of the AMOC occurs in the middle of the glacial termination, $CO_2$ concentration continues to rise during interglacial, similar to Holocene. We also discuss potential contribution of the brine rejection mechanism for the $CO_2$ and carbon isotopes in the atmosphere and the ocean during the past glacial termination.

## 1. Introduction

Antarctic ice cores reveal that during the past 800 kyr, the atmospheric $CO_2$ concentration (Petit et al., 1999; Jouzel et al., 2007) varied synchronously with the global ice volume (Waelbroeck et al., 2002; Spratt and Lisiecki, 2016). The most

straightforward explanation for this fact is that $CO_2$ drives glacial cycles together with orbital variations, and the longest, 100-kyr component of the late Quaternary glacial cycles, which is absent in the orbital forcing, is the direct response to $CO_2$ forcing where 100-kyr component is the dominant one. However, simulations with climate-ice sheet models of different complexity (e.g. Berger et al., 1999; Crowley and Hyde, 2008; Ganopolski and Calov, 2011; Abe-Ouchi et al., 2013) show that long glacial cycles (i.e. cycles with typical periodicity of ca. 100 ka) can be simulated with constant $CO_2$ concentration if the latter is sufficiently low. Moreover, these model simulations show that not only the dominant periodicity, but also the timing of glacial cycles, can be correctly simulated without $CO_2$ forcing. This fact strongly suggests an opposite interpretation of close correlation between global ice volume and $CO_2$ during Quaternary glacial cycles – namely that glacial cycles represent a strongly nonlinear response of the Earth system to orbital forcing (Paillard, 1998) while variations in $CO_2$ concentration are directly driven by ice sheets fluctuations. In turn, $CO_2$ variations additionally strongly amplify and globalize the direct response of ice sheets to the orbital forcing.

In spite of significant number of studies aimed to explain low glacial $CO_2$ concentrations (e.g. Archer et al., 2000; Sigman and Boyle, 2000; Watson et al., 2000), the influence of ice sheets on carbon cycle remains poorly understood. It is also unclear how much of $CO_2$ variations represent the direct response to ice sheets forcing and how much is the results of additional amplification of $CO_2$ variations through the climate-carbon cycle feedback. Indeed, although radiative forcing of ice sheets contributes about a half to glacial–interglacial variations in global temperature (Brady et al., 2013), most of cooling associated with ice sheets is restricted to the area covered by ice sheets and their close proximity. Thus the direct contribution of ice sheets to glacial ocean cooling is rather limited and therefore the effect of ice sheets on $CO_2$ drawdown through the solubility effect can explain only a fraction of reduction in glacial $CO_2$. At the same time, the direct effect of ice sheets on atmospheric $CO_2$ concentration through ca. 3% changes in the ocean volume and global salinity is rather well understood but works in the opposite direction and leads to glacial $CO_2$ rise of about 10-20 ppm (Sigman et al., 2000; Brovkin et al., 2007). Another direct effect of ice sheet growth on the carbon cycle through reducing area covered by forest (e.g. Prentice et al., 2011) also operates in the opposite direction. However, several other processes could potentially contribute to glacial $CO_2$ drawdown through ice sheets growth and related lowering of sea level. One such mechanism is enhanced biological productivity in the Southern Ocean due to the iron fertilization effect (Martin, 1990; Watson et al., 2000). The latter is attributed to enhanced dust deposition over the Southern Ocean seen in the paleoclimate records (Martinez-Garcia et al., 2014; Wolff et al., 2006). At least part of this enhanced deposition is associated with the dust mobilization from exposed Patagonian shelf and glaciogenic dust production related to Patagonian ice cap (Mahowald et al., 1999; Sugden et al., 2009). A number of studies on the effect of iron fertilization suggested a contribution of 10 to 30 ppm to the glacial $CO_2$ decrease (e.g. Watson et al., 2000; Brovkin et al., 2007). Another effect is related to the brines rejection mechanism, more specifically, to a much deeper penetration of brines produced during sea ice formation in the Southern Ocean during glacial time. The latter is explained by shallowing and significant reduction of the Antarctic shelf area. According to Bouttes et al. (2010) this mechanism, in combination with enhanced stratification of the deep ocean, can contribute up to 40 ppm to the glacial $CO_2$ lowering.

Apart from the mechanisms mentioned above, many other processes have been proposed to explain low glacial $CO_2$ concentration. Among them are changes in the ocean circulation (Watson et al., 2015) and an increase in the South Ocean stratification (e.g. Kobayashi et al., 2015), increase in sea ice area in Southern Ocean (Stephens and Keeling, 2000) and a shift in the westerlies (Toggweiler et al., 2006), increase in nutrients inventory or change in the marine biota stoichiometry (Sigman et al., 2000; Wallmann et al., 2016), changes in coral reefs accumulation and dissolution (Opdyke and Walker, 1992), accumulation of carbon in the permafrost regions (Ciais et al., 2012; Brovkin et al., 2016), variable volcanic outgassing (Huybers and Langmuir, 2009) and several other mechanisms. Most of these processes are not directly related to the ice sheets area or volume, and thus should be considered as amplifiers or modifiers of the direct response of $CO_2$ to ice sheets operating through the climate-carbon cycle feedbacks. Although paleoclimate records provide some useful constraints, the relative role of particular mechanisms at different stages of glacial cycles remains poorly understood.

Most studies of glacial-interglacial $CO_2$ variations performed up to date were aimed at explanation of low $CO_2$ concentration at the Last Glacial Maximum (LGM, ca. 21 ka). In these studies, both continental ice sheets and the radiative forcing of low glacial $CO_2$ concentration were prescribed from paleoclimate reconstructions. Only few attempted to explain $CO_2$ dynamics during part (usually glacial termination) or the entire last glacial cycle with models of varying complexity from simple box-type models (e.g. Köhler et al., 2010; Wallmann et al., 2016), models of intermediate complexity (Brovkin et al., 2012; Menviel et al., 2012), or stand-alone complex ocean carbon cycle models (Heinze et al., 2016). In all these studies, radiative forcing of $CO_2$ (or total GHGs) was prescribed based on paleoclimate reconstructions. Similarly, ice sheets distribution and elevation were prescribed from paleoclimate reconstructions or model simulations where radiative forcing of GHGs has been prescribed. Thus in all these studies, $CO_2$ was treated as an external forcing rather than an internal feedback. Here we for the first time performed simulations of the Earth system dynamics during the past four glacial cycles using fully interactive ice sheet and carbon cycle modelling components, and therefore the only prescribed forcing in this experiment is the orbital forcing.

## 2. The model and experimental setup

### 2.1 CLIMBER-2 model description

In this study we used the Earth system model of intermediate complexity CLIMBER-2 (Petoukhov et al., 2000; Ganopolski et al., 2001). CLIMBER-2 includes a 2.5-dimensional statistical-dynamical atmosphere model, a 3-basin zonally averaged ocean model coupled to a thermodynamic sea ice model, the 3-dimensional thermomechanical ice sheet model SICOPOLIS (Greve, 1997), the dynamic model of the terrestrial vegetation VECODE (Brovkin et al., 1997) and the global carbon cycle model (Brovkin et al., 2002, 2007). Atmosphere and ice sheets are coupled bi-directionally using a physically based energy balance approach (Calov et al., 2005). The ice sheet model is only applied to the Northern Hemisphere. The

contribution of the Antarctic ice sheet to global ice volume change is assumed to be constant during glacial cycles and equal to 10%. The model also includes parameterization of the impact of aeolian dust deposition on snow albedo (Calov et al., 2005; Ganopolski et al., 2010). The CLIMBER-2 model in different configurations has been used for numerous studies of past and future climates, in particular, simulations of glacial cycles (Ganopolski et al., 2010; Ganopolski and Calov, 2011; Willeit et al., 2015; Ganopolski et al., 2016) and carbon cycle operation during the last glacial cycle (Brovkin et al., 2012).

As it has been shown in Ganopolski and Roche (2009), temporal dynamics of the Atlantic meridional overturning circulation (AMOC) during glacial terminations in CLIMBER-2 are very sensitive to the magnitude of freshwater flux to the North Atlantic. To explore different possible deglaciation evolutions, together with the standard model version, we performed an additional suit of simulations where the component of freshwater flux into the ocean originated from melting of ice sheets was uniformly scaled up or down by up to 10%. This rather small change in the freshwater forcing (typically smaller than 0.02 Sv) does not affect AMOC dynamic appreciably during most of time but does induce a strong impact during deglaciations (see below). Other modifications of the climate-ice sheet component of the model are described in the Appendix.

The ocean carbon cycle model includes modules for marine biota, oceanic biogeochemistry, and deep ocean sediments. Biological processes in the euphotic zone (the upper 100 m in the model) are explicitly resolved using the model for plankton dynamics by Six and Maier-Reimer (1996). The sediment diagenesis model (Archer, 1996; Brovkin et al., 2007) calculates burial of $CaCO_3$ in the deep sea, while shallow-water $CaCO_3$ sedimentation is simulated based on the coral reef model (Kleypas, 1997) driven by sea level change. Silicate and carbonate weathering rates are scaled to the runoff from the land surface; they are also affected by sea level change (Munhoven, 2002). Compare to Brovkin et al (2012) the carbon cycle model has been modified in several aspects. Similar to Brovkin et al. (2012), the efficiency of nutrients utilization in the Southern Ocean is set to be proportional to the dust deposition rate (see Appendix), which in the case of one-way coupling is prescribed to be proportional to the dust deposition in EPICA ice core. However, in the fully coupled experiment, the dust deposition rate over the Southern Ocean has been computed from simulated sea level (see Appendix). This means that in the fully interactive run (see below) we did not use explicitly any paleoclimate data to drive the model and the orbital forcing was the only driver of the Earth system dynamics. In the marine carbon cycle component, we also account for a dependence of the remineralization depth on ocean temperature following Segschneider and Bendtsen (2013) (see Appendix). In the previous studies, remineralization depth was kept constant.

The CLIMBER-2 model used in earlier studies of glacial carbon cycle did not include long-term terrestrial carbon pools such as permafrost carbon, peat and carbon buried beneath the ice sheets. In the present version of the model these pools are included. The model also accounts for peat accumulation. Modification of the terrestrial carbon cycle components is described in detail in the Appendix. For simulation of atmospheric radiocarbon during the last glacial termination we used the rate of [14]C production following scenario by Hain et al. (2014) which is based on the production model by Kovaltsov et al. (2012).

## 2.2 One-way coupled and fully interactive experiments

In our previous experiments performed with the CLIMBER-2 model (Brovkin et al., 2012; Ganopolski and Brovkin, 2015) we not only prescribed temporal variations in the Earth's astronomical parameters (eccentricity, precession and obliquity) but also the radiative effect of GHGs ($CO_2$, $CH_4$ and $N_2O$) computed using their concentrations from the ice cores records (Luthi et al., 2008; Petit et al., 1999). In these experiments, which we will denote hereafter as "one-way coupled" (Fig. 1a, Table 1), atmospheric $CO_2$ was computed by the carbon cycle module but not used as the radiative forcing for the climate component. Similarly, in these experiments $CO_2$ fertilization effect on vegetation was computed using reconstructed $CO_2$ concentration. Therefore in one-way coupled experiments there were no feedbacks of the simulated atmospheric $CO_2$ concentration to climate. In the present study, we performed a suit of one-way coupled experiments for the last four glacial cycles but we also performed fully interactive simulations in which the orbital forcing was the only prescribed external forcing. Since CLIMBER-2 does not include methane and $N_2O$ cycles and does not account for these GHGs in its radiative scheme, we made use of the fact that $CO_2$ is the dominant GHG and that temporal variations of other two follow rather closely $CO_2$. To account for the effect of methane and $N_2O$ forcings, we computed the effective $CO_2$ concentration used in the radiative scheme of the model in such a way that radiative forcing of equivalent $CO_2$ exceeds radiative forcing of simulated $CO_2$ by 30% at any time. This type of experiments we will refer to as "fully interactive" (Fig. 1b). In the fully interactive experiment we use computed $CO_2$ concentration also in terrestrial component to account for $CO_2$ fertilization effect. As was stated above, dust deposition over the Southern Ocean used in the parameterization of iron fertilization effect computed from the global sea level. The radiative forcing of aeolian dust and dust deposition on ice sheets (apart from the glaciogenic dust sources) in both types of experiments were obtained identically to Calov et al. (2005) and Ganopolski and Calov (2011) by scaling the field computed with GCMs, where scaling parameter was proportional to global ice volume.

## 2.3 Model spin-up

The model spin-up and proper choice of model parameters for simulation of multiple glacial cycles represents a challenge when using the models with very long-term components of the carbon cycle because inconsistent initial conditions or even a small disbalance in carbon fluxes could lead to a large drift in simulated atmospheric $CO_2$ concentration (in the case of one-way coupling) or the state of the entire Earth system (climate, ice sheets, $CO_2$) in the case of fully interactive experiments. Note that in the latter case, the negative climate-weathering feedback will eventually stabilize the system but this occurs at the time scale of several glacial cycles and over this time climate could drift far away from its realistic state. To avoid such drift, volcanic outgassing should be carefully calibrated. Based on a set of sensitivity experiments, we found that the value of 5.3 Tmol C/yr allows us to simulate quasiperiodic cycles without long-term trend in atmospheric $CO_2$. Note, that even ±10% change in volcanic outgassing leads to significant (order of 100 ppm) drift in $CO_2$ concentration simulated over the last four glacial cycles.

When the carbon cycle model incorporates such long-term processes as terrestrial weathering, marine sediment accumulation and permafrost carbon burial, the assumption that the system is close to equilibrium at preindustrial period or at any other moment of time is not valid even if $CO_2$ concentration was relatively stable during a certain time interval. To produce proper initial conditions at 410 ka we performed a sequence of 410 ky-long one-way coupled runs with the identical forcings. We first used as the initial conditions the final state obtained in simulation of the last glacial cycles (Brovkin et al., 2012). Then we launched each 410 ky experiment from the final state obtained in the previous model run. The results of such sequence of experiments reveal a clear tendency to converge to the solution with similar initial and final states of the Earth system. We then used the state of climate and carbon cycle obtained at the end of the last run as the initial conditions for all experiments presented in this paper. In the analysis of all experiments described below we exclude the first 10,000 years when adaptation of different fields to each other occurs.

## 3. Simulations of the last four glacial cycles

Realistic simulation of climate and carbon cycle evolution during the last four glacial cycles is more challenging in the case of fully interactive configuration, because in this case a number of additional positive feedbacks tend to amplify initial model biases. Therefore we begin our analysis with the one-way coupled simulations similar to that performed in Brovkin et al. (2012). This configuration was also used for calibration of new parameterizations (see section 4) and sensitivity experiments for the last glacial termination (section 5).

### 3.1 Experiments with one-way coupled climate-carbon cycle model

Simulated climate and ice sheets evolution in the one-way coupled experiments are rather similar to the ones in Ganopolski and Calov (2011), which is not surprising since the only difference between model versions used in these studies is related to the coupling between ice sheet and climate components (see Appendix). Simulated glacial cycles are characterized by global surface air temperature variations of about 5$^{\circ}$C (not shown) and maximum sea level drops by more than 100 meters during several glacial maxima. Simulated global ice sheets volume during most of time is close to the reconstructed one (Spratt and Lisiecki, 2016) (Fig. 2d). In general, differences between simulated and reconstructed global sea level are comparable or smaller than uncertainties in sea level reconstructions obtained using different methods.

Simulated $CO_2$ concentration (Fig. 2e) is also in a good agreement with reconstructions based on several Antarctica ice cores (Barnola et al., 1987; Monnin et al., 2004; Petit et al., 1999; Luthi et al., 2008). The model correctly reproduces the magnitude of glacial-interglacial $CO_2$ variability of about 80 ppm. Results of simulations with the standard model version (ONE_1.0) and model with 10% enhanced meltwater flux (ONE_1.1) are essentially identical during most of time except for glacial terminations. During glacial terminations even rather small differences in the freshwater forcing cause pronounced differences in the temporal evolution of the AMOC, and as a result, of $CO_2$ concentration. As seen in Fig. 2d, in the

experiment ONE_1.0, $CO_2$ concentration grows monotonously during the last glacial termination (TI, midpoint at ca. 15 ka) and TIV (ca. 330 ka) while it rises faster and overshoots the interglacial level during TII (ca. 135 ka) and TIII (ca. 240 ka). To the contrary, in the experiments ONE_1.1, similar overshoots occur during TI and III but not TIV. In all cases, simulated $CO_2$ lags behind the reconstructed one but this lag is smaller in the case when overshoot is simulated. Experiments with $CO_2$

overshoots are clearly in better agreement with empirical data for MIS7 and MIS9. Analysis of model results shows that pronounced $CO_2$ overshoot occurs in the case when the AMOC is suppressed during the entire glacial termination and recovers only after the cessation of the meltwater flux (Fig. 3). To the contrary, if the AMOC recovers well before the end of deglaciation, simulated $CO_2$ experiences only local overshoot and continues to rise during most of the interglacial. The latter behaviour is similar to that was observed during MIS 11 and the Holocene, while the former is typical for MIS5, 7 and 9.

Thus our model is able to reproduce both types of $CO_2$ dynamics during the interglacials.

The rise of $CO_2$ by 10-20 ppm on millennial time scale during the AMOC shutdowns is the persistent feature of CLIMBER-2 and the cause of this rise has been explained in Brovkin et al. (2012) by a weakening of the reverse cell of the Indo-Pacific overturning circulation during periods of reduced AMOC. A similar rise in atmospheric $CO_2$ concentration during periods of AMOC shutdown has been simulated in some other (but not all) similar modeling experiments.

Incorporation of the temperature-dependent remineralization depth additionally contributes to the $CO_2$ overshoots at the beginning of several interglacials (see below) but the mechanism described in Brovkin et al. (2012) remains the dominant one.

Comparison of simulated deep ocean $\delta^{13}C$ with paleoclimate reconstructions (Fig. 4) show that the model correctly simulates larger $\delta^{13}C$ variability in the deep Atlantic in comparison to the deep Pacific but underestimates the amplitude of

glacial-interglacial $\delta^{13}C$ variability. Simulated atmospheric $\delta^{13}CO_2$ shows a rather complex behaviour and amplitude of variability up to 0.6‰. The agreement between the simulated and reconstructed (Eggleston et al., 2016) atmospheric $\delta^{13}CO_2$ is rather poor. Both model and data show a drop in atmospheric $\delta^{13}CO_2$ during the last and penultimate deglaciations but the data suggest also the strong drop at the end of Eemian interglacial while the model simulated continuous rise of $\delta^{13}CO_2$ at that interval. In addition, temporal variability of the reconstructed $\delta^{13}CO_2$ is significantly larger than the simulated one.

More detailed comparison with empirical data during the last deglaciation is presented in the Section 5.

Changes in the ocean oxygenation is considered to be an important indicator of respired carbon storage in the deep ocean, and therefore the proxy for the strength of ocean biological pump. Jaccard et al. (2016) inferred a significant decline in the deep South Ocean oxygenation and interpreted it as the result of combine effect of iron fertilization by dust and decreased deep ocean ventilation. Our results (Fig. 5) are fully consistent with such interpretation. The model simulates significant

reduction of the dissolved oxygen in the deep South Ocean during glacial period. Roughly 2/3 of this reduction is simulated already in the experiment without iron fertilisation and can be solely attributed to the reduced deep ocean ventilation. It is noteworthy that changes in the oxygen concentration in this experiment are strongly anticorrelated with the area of sea ice in the South Hemisphere (Fig. 5c). The late is explained by the fact that sea ice directly and indirectly (through stratification of

the upper ocean layer) affects gas exchanges between the ocean and the atmosphere. Oxygen concentration is additionally reduced in the experiment which accounts for the iron fertilization effect during period with high dust deposition rate (Fig. 5d).

## 3.2. Experiments with the fully interactive model

In two-way coupling experiments (fully interactive runs), orbital forcing is the only prescribed forcing and the model does not use any time-dependent paleoclimatological information (such as the Antarctic dust deposition rate used in the one-way coupled experiment). Results of fully interactive experiment INTER_1.0 are shown in Fig. 6. For the first experiment of this type ever, the agreement between model simulations and empirical reconstructions is reasonably good. The model simulates correct magnitude and timing of the last four glacial cycles both in respect of sea level and $CO_2$ concentration. It also reproduces strong asymmetry of glacial cycles. Naturally, the mismatch between simulated and reconstructed characteristic in fully interactive experiments is larger than in the one-way coupled experiment. In particular, in the fully interactive experiment, simulated ice volume is underestimated by 10-20 meters compared to reconstructed one. Although the magnitude of glacial-interglacial $CO_2$ variability in the fully interactive experiment INTER_1.0 is similar to that in one-way coupled experiment ONE_1.0 and in reconstructions, the lag between simulated and reconstructed $CO_2$ during glacial terminations increases additionally in comparison to one-way coupled experiment. Interestingly, the last glacial cycle and the first 150 ky of the INTER_1.0 and ONE_1.0 runs are in very good agreement while during time interval between 300 ka and 150 ka BP discrepancies are larger. This period corresponds to higher eccentricity and therefore larger magnitude of the orbital forcing. Similarly to the results of one-way coupled experiments, fully interactive runs also show strong sensitivity to magnitude of freshwater flux during glacial terminations.

Comparison of simulated ice sheets spatial distribution and elevation (Fig. 7) shows that the results of one-way coupled (ONE_1.0, Fig 6a) and fully interactive run (INTER_1.0, Fig. 7b) are almost identical during the LGM (the same is true for the previous glacial maxima, not shown) and in a reasonable agreement with the paleoclimate reconstructions. During glacial terminations, the difference between two runs increases since in the fully interactive run the radiative forcing of GHGs lags considerably behind the reconstructed one used in the one-way coupled experiment. As the result at 7 ka continental ice sheets melted completely in the one-way coupled experiment (Fig. 7c) while in the fully interactive run a relatively large ice sheet is still present in the northern-eastern Canada (Fig. 7d).

It is instructive to compare frequency spectra of simulated and reconstructed global ice volume in one-way and fully coupled experiments (Fig. 8. In addition, we show here results from the experiment ONE_240 performed with constant radiative forcing of GHGs corresponding to equivalent $CO_2$ concentration of 240 ppm. As already shown by Ganopolski et al. (2011), even with constant $CO_2$, the model computes pronounced glacial cycle with 100-kyr periodicity, although it has much weaker amplitude than the reconstructed sea level. Both model experiments with varying $CO_2$ radiative forcing

(ONE_1.0 and INTER_1.0) reveal much stronger 100-kyr periodicity, which has only slightly weaker amplitude than the spectrum of reconstructed sea level. Interestingly, frequency spectra of sea level simulated in one-way and fully interactive runs have rather similar power in 100 ka and obliquity (40 ka) bands, but in the precessional band (ca. 20 ky) one-way coupled experiment reveals much higher spectral power. This cannot be explained by the prescribed radiative forcing of GHGs because the latter contains very little precessional variability. The explanation of stronger precessional component in the ONE_1.0 run is related to the fact that one-way coupled model simulates slightly faster ice sheet growth during the initial part of each glacial cycle and the modeled sea level variability at the precessional frequency is very sensitive to ice volume.

## 4. The composition of "the carbon stew" and factor analysis

In this section, we discuss the contribution of different factors to simulated variations in $CO_2$ concentration. Because neither of mechanisms could explain the $CO_2$ dynamics in isolation from the other factors (e.g. Sigman and Boyle, 2000; Archer et al., 2000), we call the composition and timing of the mechanisms leading to the glacial $CO_2$ cycle "the carbon stew". As has been shown in Brovkin (2012), the role of different mechanism controlling $CO_2$ concentration at different phases of glacial cycles is different. However, even if we consider only the LGM (as most of previous work did), the composition of "carbon stew" remains highly uncertain even although there is a growing awareness that both physical and bilological processes must have played a comparably important role in glacial $CO_2$ drawdown (e.g. Schmittner and Somes 2016; Galbraith and Jaccard, 2015). Obviously, the choice of the "carbon stew" is crucially important for successful simulations of glacial cycles. The aim of our paper is not to present the ultimate solution for the "carbon stew" problem since at present this is impossible. Rather we want to demonstrate that with a reasonable representation of physical, geochemical and biological processes in the model, it is possible to reproduce the main features of Earth system dynamics over the past 400 kyr, including the magnitude and timing of climate, ice volume and $CO_2$ variations.

Similar to the study by Brovkin et al. (2012), we performed a set of experiments using one-way coupling (see Table 1 for detail). We use this approach instead of fully interactive coupling to exclude complex and strongly nonlinear interactions associated with ice sheet dynamics which significantly complicate factor analysis. In the case of one-way coupled experiments climate, ice sheets and other external factors are identical and experiments only differ by parameters of the carbon cycle model. Since $CO_2$ simulated in the one-way coupled experiment with 10% enhanced meltwater flux (ONE_1.1) is in a slightly better agreement with observational data than the standard one (ONE_1.0), for the factor analysis we used experiment ONE_1.1 as the reference one and performed all sensitivity experiments with 10% enhanced meltwater flux.

### 4.1 The standard carbon cycle model setup

We begin our analysis from the experiment that incorporates only standard ocean biogeochemistry as described in Brovkin et al. (2007) (Fig. 9). This experiment does not include effect of terrestrial carbon cycle. In this configuration, the model is able

to explain only about 45 ppm of $CO_2$ reduction during glacial cycles. Note that this experiment accounts for changes in the ocean volume by ca. 3% and corresponding changes in the total biogeochemical inventories including salinity. These volume changes are often neglected in simulations with 3-dimensional ocean models (e.g. Heinze et al. 2016), although in our simulations they counteract to glacial $CO_2$ drawdown by ca. 12 ppm. Without the effect of ocean volume reduction, the combination of physical processes and carbonate chemistry can explain of up to 57 ppm at the LGM and 38 ppm during the entire 400 kyr time interval (see Table 2). This is consistent with the resent results by Buchanam et al. (2016) and Kobayashi et al. (2015). Note that simulated changes in silicate weathering and its impact on atmospheric $CO_2$ are small as have been shown already in Brovkin et al. (2012).

Accounting for the land carbon changes does not help to explain the $CO_2$ concentration changes, since terrestrial carbon contains by ca. 350 Gt less carbon at the LGM compared to the pre-industrial state. This reduces the glacial-interglacial $CO_2$ difference by 10-15 ppm comparing to the ocean-only experiment (Fig. 9b). Enabling of parameterization for the iron fertilization effect in the Southern Ocean results in additional glacial $CO_2$ drawdown of up to 30 ppm (22 ppm at the LGM), mostly towards the end of each glacial cycle which is related to the chosen parameterization for the dust deposition rate (Fig. 9c). This value is rather close to that reported by Lambert et al. (2015). With all these processes considered in our previous study by Brovkin et al. (2012), we are still short of ca. 25 ppm to explain the full magnitude of glacial-interglacial variability.

## 4.2 Additional processes included in the carbon cycle model

There is a number of other proposed mechanisms which can explain several tens ppm of glacial $CO_2$ decline. Our choice of two processes to obtain the observed magnitude of glacial-interglacial $CO_2$ variations is somewhat subjective. Chosen mechanisms are explained below, while an alternative one (brine rejection) is discussed in the section 4.3.

The first additional to Brovkin et al. (2012) mechanism is temperature-dependent remineralization depth. In the standard CLIMBER-2 version, remineralization depth is spatially and temporally constant. Since in the colder ocean remineralization depth increases, this enhances the efficiency of carbon pump and contributes to a decrease of atmospheric $CO_2$ concentration (e.g. Heinze et al., 2016; Menviel et al., 2012; Matsumoto, 2007). Details of the mechanism implementation are described in Appendix. As seen from Fig. 9e, making remineralization depth temperature-dependent introduces additional glacial-interglacial variability with the magnitude of about 20 ppm. Roughly half of this value is clearly attributed to the $CO_2$ overshoots which are seen at the beginning of some interglacials. The reason is that the AMOC shutdowns due to melt water flux that happened during glacial terminations lead not only to surface cooling in the North Atlantic, but also to significant thermocline warming that occurs over the entire Atlantic ocean (e.g. Mignot et al., 2007). This subsurface warming causes significant shoaling of the remineralization depth and the release of carbon from the ocean into the atmosphere. This process reverses after the recovery of the AMOC at the beginning of interglacials.

Burley and Katz (2015) and Huybers and Langmuir (2009) proposed that the rate of volcanic outgassing varies during glacial cycle due to variable load of the ice sheet and ocean on the Earth crust. Therefore we assume that volcanic outgassing has a variable component (about 30% of its averaged value of 5.3 Tmol/yr) which represent the delayed response to the change in ice volume. This simple parameterization explained in Appendix does not affect cumulative volcanic outgassing over glacial cycle, but contributes to glacial-interglacial variability by additional 10 ppm (Fig. 9d). With varying volcanic outgassing and temperature-dependent mineralization depth, CLIMBER-2 model reproduces glacial-interglacial $CO_2$ cycles in a good agreement with paleoclimate records (Fig. 9a).

## 4.3 Brine rejection mechanism

Using a different version of CLIMBER-2, Bouttes et al. (2010) proposed that a significant fraction of glacial-interglacial $CO_2$ variations can be explained by the mechanism of brine rejections, more specifically, by a large increase in the depth to which brines can penetrate under glacial conditions without significant mixing with ambient water masses. Such increase in brine efficiency under glacial conditions would result in large transport of salinity, carbon and other tracers from the upper ocean layer into the deep ocean. By choosing the efficiency coefficient close to one, Bouttes et al. (2010) demonstrated that brines are able to explain up to 40 ppm $CO_2$ decrease. We have implemented this mechanism in combination with stratification-dependent vertical diffusivity in our version of the CLIMBER-2 model and got results qualitatively similar to Bouttes et al. (2010).

While we think that the brine rejection mechanism belongs to a class of plausible mechanisms contributing to glacial $CO_2$ drawdown, we did not use brine parameterization in our simulations for several reasons. Firstly, the parameterization for brine rejection cannot be tested against observational data. For present day climate conditions, brine rejections efficiency should be below 0.1, otherwise modern Antarctic bottom water becomes saltier than the North Atlantic deep water which is at odds with reality. This means that to be an efficient mechanisms for glacial $CO_2$ drawdown, the brine efficiency should increase under glacial conditions at least by an order of magnitude. Whether this is physically plausible is not clear. The only paleoclimate constraint on the brine efficiency is reconstruction of paleosalinity based on the pore water (Adkins, et al. 2002) which suggests increase of deep water salinity in the Southern Ocean by more than 2 psu during the LGM. Such increase in salinity is indeed difficult to reproduce without contribution of brines However the accuracy of salinity reconstruction based on such method remains uncertain (Wunsch, 2016). Second, there is a problem with temporal dynamics of brine rejection efficiency. Mariotti et al. (2016) assumed abrupt decrease of brine rejection efficiency from 0.7 to 0 in a very short interval between 18 and 16 ka. However, both sea level and the size of Antarctic ice sheets were essentially constant during this period and therefore there is no obvious reason for such large variations in the brine rejection efficiency. According to the interpretation of Roberts et al. (2016), brines rejection remained efficient during most of glacial termination and ceased only after 11 ka when most of glacial-interglacial $CO_2$ rise has been already accomplished. In the view of these uncertainties, we decided not to include parameterizations of brine rejection mechanism in simulations of glacial cycles. However, for

simulations of the last glacial termination discussed below, we analysed potential effect of brine rejection on radiocarbon and other paleoclimate proxies.

## 5. Simulations of the Termination I

### 5.1 Simulation of climate, $CO_2$, and carbon isotopes during the last termination

The last glacial termination provides a wealth of paleoclimate records with a potential to better constrain the mechanisms of glacial $CO_2$ variability. In this section, we discuss the last glacial terminations in more detail. Similarly to the previous
section, to exclude nonlinear interaction with ice sheets, we discuss here only one-way coupled experiments. To reduce computational time, we performed experiments only for the last 130,000 years starting from the Eemian interglacial and using the same initial conditions as in the experiments discussed above.

In the standard ONE_1.0_130K experiment, the model simulates climate variability across the Termination I rather realistically. In particular, it reproduces temporal resumption of the AMOC in the middle of the termination resembling
Bølling-Allerød warm event (Fig. 10a). The timing of this event in our model is shifted by ca 1000 years compared to the paleoclimate records. Results of our experiments reveal a high sensitivity of the timing of the AMOC resumption to the magnitude of freshwater flux. A change of the flux by just 2% in the ONE_0.98_130K experiment significantly alters millennial scale variability during the last glacial termination (Fig. 10). This result suggests that simulated millennial scale variability during the Termination I is not robust, i.e. it is unlikely that a single model run through the glacial termination
would reproduce the right timing or even the right sequence of millennial-scale events.

Although simulated $CO_2$ concentration at LGM and pre-industrial state are close to observations, simulated $CO_2$ appreciably lags behind reconstructed $CO_2$ during the termination (Fig. 10b). This is primarily related to the fact that simulated $CO_2$ does not start to grow at ca. 18 ka BP as reconstructed, but only after the end of simulated analogue of Bølling-Allerød event. At the same time, in agreement with paleoclimate reconstructions, $CO_2$ concentration reaches a local
maximum at the end of the North Atlantic cold event, which resembles the Younger Dryas. Simulated $CO_2$ concentration also reveals continuous $CO_2$ rise during Holocene towards preindustrial value of 280 ppm. This result confirm that such $CO_2$ dynamics could be explained by only natural mechanisms and does not require early anthropogenic $CO_2$ emissions until ca. 2 ka (Kleinen et al., 2016). This result also demonstrates that temporal dynamics of $CO_2$ during interglacials critically depends on the timing of final AMOC recovering. Late recovery during glacial termination causes strong overshoot of $CO_2$ at the
beginning of interglacial following by some decrease or stable $CO_2$ concentration. However, if the complete AMOC recovery occurs well before the end of termination, only temporal $CO_2$ overshoot occurs and $CO_2$ continues to rise during the entire interglacial.

It is assumed that atmospheric $\delta^{13}C$ provides useful constraint on the mechanisms of deglacial $CO_2$ rise (Schmitt et al., 2012; Joos et al., 2004; Fischer et al., 2010). Simulated atmospheric $\delta^{13}C$ drops from the LGM level of about -6.4‰ to the

minimum value of -6.7‰ between 16 and 14 ka (Fig. 10c). This is primarily related to the reduction of marine biological productivity which, in turn, is explained by the decrease of iron fertilization effect over the Southern Ocean during the first part of Termination I. The magnitude of the $\delta^{13}C$ drop is in a good agreement with empirical data (Fig. 10c). The model is also able to simulate W-shaped $\delta^{13}C$ evolution associated with reorganization of the AMOC, however, this W-shape is

shifted in time comparing to the reconstructed one by ca. 1000 years because model analogue of Bølling-Allerød event occurs earlier than the real one by the same amount of time. Note that this local maximum in $\delta^{13}C$ is completely absent in the experiment ONE_1.1_130K where temporal resumption of the AMOC during glacial termination does not occur. $\delta^{13}C$ rise after 12 ka is primary attributed to the accumulation of carbon in terrestrial carbon pools (forest regrowth and peat accumulation). At the same time, simulated present-day atmospheric $\delta^{13}C$ is underestimated compared to ice-core data by ca.

0.15‰.

   The model simulates almost monotonous decrease of atmospheric $\Delta^{14}C$ from the LGM to present. Most of this decrease (ca. 200‰) is caused by prescribed production rate which was about 20% higher during LGM. Only about 80‰ of $\Delta^{14}C$ is attributed to difference in climate state between LGM and present, primarily, due to less ventilated deep ocean. As shown in Fig. 10d, simulated atmospheric $\Delta^{14}C$ is significantly underestimated before 12 ka compared to reconstruction by Reimer et

al. (2013) and at LGM this difference reaches more than 100‰. It is possible but unlikely that such big differences can be attributed to uncertainties in reconstructed production rate. An alternative hypothesis for explaining this mismatch is discussed below.

   Fig. 11 shows the LGM time slice anomalies and temporal evolution of $\delta^{13}C$ and radiocarbon ventilation age during the Termination I in the Atlantic ocean simulated in the experiment ONE_1.0_130K. Spatial distribution of glacial anomalies

and temporal dynamics of $\delta^{13}C$ and radiocarbon ventilation age during termination are qualitatively very similar. Glacial $\delta^{13}C$ in the deep Atlantic at the LGM is by 0.6-1‰ lower than at present; that is primarily related to a shoaling of the AMOC and reduced ventilation in the Southern Ocean. The vertical distribution of $\delta^{13}C$ anomalies at the LGM is consistent with the paleoclimate reconstructions (e.g. Hesse et al., 2011).

   Simulated ventilation age at the LGM can be directly compared with Skinner et al. (2017) (their Fig. 4a, c). Both models

and data show significant increase of radiocarbon ventilation age in the deep Atlantic. However, the spatial patterns of ventilation age changes are rather different. In the model, the largest increase in the ventilation age occurs in the deep Northern Atlantic which is explained by a shoaling of the AMOC cell and an increased presence of the poorly ventilated Southern Ocean water masses. At the same time, the data for the deep North Atlantic are characterised by very large scattering (from 1000 to 3000 $^{14}C$ years) and it is unclear whether their average values can be directly compared to the

results of the zonally averaged ocean model.

   During glacial termination, both $\delta^{13}C$ and radiocarbon ventilation age show pronounced response at all depth to the millennial scale reorganizations of the AMOC (Fig. 12,b,d). The ventilation age in the deep Atlantic, which is about 2000 years prior to the model analogue of warm Bølling-Allerød event rapidly reaches nearly modern level after the AMOC

resumption and drops again to glacial level during model analogue of the cold Younger Dryas event. Such evolution of ventilation age in the North Atlantic is in a good agreement with paleoclimate reconstructions (Robinson et al., 2005; Skinner et al., 2014).

## 5.2 Brine rejection mechanisms and radiocarbon in the ocean and atmosphere

As discussed above, our version of the CLIMBER-2 model is not able to reproduce accurately atmospheric [14]C decline during the first part of glacial termination. At the same time, Mariotti et al. (2016) demonstrated that their version of CLIMBER-2, which incorporates mechanism of brine rejection, is able to simulate larger atmospheric [14]C decrease from LGM till present, consistently with observational data (Reimer et al., 2013). By introducing similar parameterization for brine rejection and stratification-dependent vertical diffusivity in our model, we are able to reproduce results similar to Mariotti et al. (2016) (Fig. 12). It is noteworthy that we use different temporal dynamics of the efficiency of brine rejections during termination. Instead of abrupt and non-monotonous changes in the brine efficiency prescribed in Mariotti et al. (2016), in the ONE_BRINE_130K experiment we assume that this efficiency is 0.75 at the LGM, 0 at present, and in between it follows global ice volume. We do not claim that this scenario is more realistic, but at least it is more consistent with the findings of Roberts et al. (2016). Fig. 11 shows that the model with brine rejection and stratification-dependent vertical diffusivity simulates atmospheric $\Delta^{14}$C in better agreement with empirical data then the standard version. This is explained by the fact that brine rejection in combination with stratification-dependent vertical mixing produces very salty and dense deep water masses which are almost completely isolated from the surface. Comparison of the vertical profiles of ventilation age (Fig. 13) with the basin averaged data from Skinner et al. (2017) shows that in the Atlantic and Pacific oceans, even the standard model version overestimates the radiocarbon ventilation age of glacial water masses. In turn, the model version with the brine rejection parameterization simulates water masses which are by 500 to 1000 years older than in the standard version. Only in the Southern Ocean the reconstructed ventilation age is consistent with both models version. As the result, the standard model version simulates ca 800 [14]C yr increase in glacial ocean ventilation age at the LGM which is in a good agreement with 689 ± 53 [14]C yr reported in Skinner et al. (2017). At the same time, the model with brine rejection simulates the increase in the ventilation age by more than 1300 [14]C yr.

Interestingly, the two model versions do not differ much in respect of simulated deep ocean $\delta^{13}$C. At last, the two model versions differ significantly in respect of the deep South Ocean salinity. Change in salinity in the standard model version is only about 1 psu which is close to the global mean salinity change due to ice sheets growth. The model version with the brine rejection parameterization simulates glacial deep South Ocean salinity of above 37 psu which is in a good agreement with the reconstruction by Adkins et al. (2002). Thus we found that including additional effects (brines and stratification-dependent diffusion) helps to bring atmospheric $\Delta^{14}$C and the deep South Ocean salinity in better agreement with available reconstructions but in expense of very old (likely to be at odds with paleoclimate data) water masses in the deep ocean. Of

course, these results are obtained with a very simplistic ocean component and it is possible that more realistic ocean models would be able to resolve this apparent contradiction.

### 5.3 Changes in the terrestrial carbon cycle

The evolution of the carbon cycle in the "offline" simulation is presented on Figure 13. The "conventional" components of the land carbon cycle (vegetation biomass, soil carbon stored in non-frozen and non-flooded environment) change between 1400 GtC during glacial maxima and 2000 GtC during interglacial peaks. Such an amplitude of 600 GtC of glacial-interglacial changes is typical for the models of the land carbon cycle without long term-components (Kaplan et al., 2002; Joos et al., 2004; Brovkin et al., 2002). However, when we account for permafrost, peat, and buried carbon, the magnitude is decreasing to 300-400 GtC. This is due to counteracting effect of the permafrost and buried carbon pools relative to the conventional components. Both these pools vary between 0 and 350 GtC and reach their maxima during glacials. The peat storage also reaches about 350 GtC, but it grows only during interglacials or warm stadials. Let us note that during glacial inceptions, while biomass and mineral soil carbon decrease, terrestrial carbon storage increases due to an increase in buried and permafrost carbon. As a result, total land carbon did not change much during the period of large ice sheet initiation.

During the last deglaciation (Fig. 14, right), the peat storages increase monotonically reaching ca. 350 GtC at pre-industrial. The conventional carbon pools increase from 1400 to 1800 GtC at the peak of interglacial (ca. 9 kyr BP), and then start to decline due to orbital forcing effect on climate in northern hemisphere. The permafrost and buried carbon pools show opposite behaviour, experiencing minimum at 10 and 5 ka, respectively, and grow afterwards. The combined effect on the total land carbon is a monotonic increase during interglacials, mostly because of peat accumulation.

### 6. Discussion and conclusions

We present here the first simulations of the last four glacial cycles with one-way and two-way coupled carbon cycle model. The model is able to reproduce the major aspects of glacial-interglacial variability of climate, ice sheets and of atmospheric $CO_2$ concentration even when driven by orbital forcing alone. These results provide strong support to the idea that long and strongly asymmetric glacial cycles of the late Quaternary represent a direct but strongly nonlinear response of the Northern Hemisphere ice sheets to the orbital forcing which, in turn, is amplified and globalized by the carbon cycle feedback.

The model simulates correct timing of the past glacial terminations in terms of ice volume while the simulated $CO_2$ concentration lags behind the reconstructed one by several thousand years. The model is also able to simulate temporal evolution of the stable carbon isotope in the ocean. At the same time, the agreement between simulated and reconstructed atmospheric $\delta^{13}C$ is rather poor. Similarly, the magnitude of simulated atmospheric $^{14}C$ decline during the last glacial termination is underestimated. Introducing the brine rejection parameterization and stratification-dependent diapycnal

diffusivity allows us to improve the agreement for atmospheric $^{14}$C but leads to unrealistically "old" glacial deep ocean water masses.

Temporal dynamics of $CO_2$ during interglacial depends strongly on the timing of the AMOC recovering during glacial termination. If the AMOC recovers only at the end of glacial termination, $CO_2$ concentration experiences the overshoot at the beginning of interglacial and then $CO_2$ declines. To the contrary, early recovery of the AMOC leads to monotonous rise of $CO_2$ during interglacials. In our simulations, millennial scale variability during the last glacial termination is very sensitive to magnitude of meltwater flux, and the sequence and timing of simulated millennial scale events are not robust even when the model is forced by prescribed radiative forcing of GHGs.

Adding new long-term carbon pools (peat, buried and permafrost carbon) decreases the amplitude of glacial-interglacial changes in the total land carbon storage. It helps to reduce an effect of terrestrial biosphere on the $CO_2$ change during glacial inception and to lesser extent during glacial terminations.

This work demonstrates that simulation of glacial cycles with Earth system models driven by orbital forcing alone is possible. This does not mean that we presented here the ultimate solution for the accurate recipes for all processes and feedbacks and, in particular, for "the carbon stew". The understanding of how global carbon cycle operates on orbital and suborbital time scales still remains incomplete and large uncertainties remain in the choice of individual processes and their parameterisations. Paleoclimate data provide some useful constrains but the proxy data syntheses are in the state far from being perfect, with some proxies telling contradicting stories and others are difficult to interpret.

The CLIMBER-2 model is rather simple and coarse resolution Erath system model. This allows us to perform a large ensemble of model simulations on orbital and even longer time scales. Obviously, such fast model has significant limitations, in particular, it employs the zonally averaged ocean model. Many essential processes, such as iron fertilization effect, are parameterized. The development of a high-resolution state-of-the-art Earth system model suitable for simulation of the interaction between climate, ice sheets and carbon cycle at the orbital time scales is absolutely crucial to make the next step forward in understanding of the Earth system dynamics during Quaternary.

**Acknowledgements**. The authors are thankful to Edouard Bard and Fortunat Joos for helpful discussion of the atmospheric and oceanic $^{14}$C dynamics and Luke Skinner for usefully comments and suggestions. The authors acknowledge support by the German Ministry of Education and Research (PalMod-Project)

**Appendixes**

**A1. Modifications of terrestrial carbon cycle model**

The old version of the CLIMBER-2 carbon cycle module described in Brovkin et al (2002) considers two vegetation types –
trees and grass. Each of two vegetation types has four carbon pools – leaves, stems, fast and slow soil carbon. Each of these
four pools occupies the same fraction of grid cell as the respective vegetation type. Crichton et al. (2014) in their version of
permafrost carbon implementation into CLIMBER-2 have not changed the pool structure but modified turnover time,
assuming that it is increasing under permafrost conditions. In the new version of the carbon cycle module which we use in
present work, we introduced three new carbon pools: boreal peat, permafrost, and carbon buried under ice sheets (Fig. A1).
The fractions of land covered by grass and trees are computed in the vegetation model following (Brovkin et al., 1997), the
fraction of land covered by ice sheets is computed by the ice sheet model and the fraction of permafrost $f_{pm}$ for the
temperature range $-5^{\mathrm{o}}\mathrm{C} < T_{ts} < 5^{\mathrm{o}}\mathrm{C}$ is computed in the land surface module as

$$f_{pm} = 0.5 - 0.1 T_{ts},$$

where $T_{ts}$ is annual mean top soil layer temperature. It is assumed that grass (in boreal latitudes this mean tundra) is located
north of forest and therefore freezes first. Only if permafrost exceeds grass fraction, the permafrost can expand over the area
covered with trees. During the ice sheet growth, all carbon under ice sheets apart from the living biomass is re-allocated into
the buried carbon pool. Buried carbon remains intact till it is covered by ice sheets. During deglaciation, this buried carbon is
transformed into the permafrost pool. Fraction of land covered by peat is define as

$$f_{pt} = f_{pt}^{*}(1 - f_{gc} - f_{pm}),$$

where $f_{pt}^{*}$ is the potential fraction of peat for each grid cell prescribed from modern observational data and $f_{gc}$ id the faction
of land covered by ice sheets. Note that we do not consider peatlands in low latitudes. Although peat and permafrost have
certain areal fractions, they are considered to be parts of grid cell covered by vegetation. Net primary production and fluxes
between the fast carbon pools (leaves, stems and fast soil pool) are computed the same way as in Brovkin et al. (2012). The
downward flux of carbon from the fast soil is partitioned between slow soil pool and permafrost proportionally to their
relative factions. The rate of peat accumulation is equal to a fixed fraction of net primary production in the respective
vegetation type. Evolution of carbon content $p_i$ in slow carbon pools is described by the equation

$$\frac{dp_i}{dt} = q_i f_i + \frac{df_i}{dt} b_i' ,$$

where $p_i = b_i f_i$, and $b_i$ is the concentration of carbon in the $i$-th carbon pool (in kgC/m$^2$), and $f_i$ is the fraction of the $i$-th pool. Value $q_i$ represents the difference between local accumulation and decay of carbon in the pool, and $b_i'$ is carbon

concentration in the pool by which $i$-th pool is expanded and $b_i' = b_i$ if $i$-th pool is shrinking. For peat $b_i' = 0$. For the permafrost, the situation is more complex, because it can gain/lost carbon from/to slow soil, peat and buried carbon pools. The source terms for the permafrost pool $q_{pm}$ consists of the sum of fluxes from the fast grassland and tree fast soil pools into the respective slow pools (see Brovkin et al., 2002 for detail) minus the decay term, where decay time scale is set to 20,000 years. Apart from carbon, terrestrial carbon model also computes carbon isotopes ($^{13}$C and $^{14}$C) contents in all carbon pools.

Since carbon isotopes are also computed in the oceanic carbon cycle model, we can compute $\delta^{13}$C and $\delta^{14}$C in the atmosphere and compare modeling results with available paleoclimate data.

**A2 Modifications of the ocean carbon cycle module**

**A2.1. Dust deposition in the Southern Ocean**

In the one-way coupled experiments, similar to Brovkin et al. (2012), we used the concentration of eolian dust in the Antarctic ice cores as the proxy for iron deposition over the Southern Ocean. Such choice is supported by recent measurements of iron content in the Southern Ocean sediments core (Lamy et al., 2014). In the fully interactive run, the iron

flux over the Southern Ocean ($D$) in arbitrary units is parameterized through the global sea level change as

$$D = (100 \frac{dS}{dt} + 10) \max(S - 50; 0) + 1.5 S ,$$

where $S$ is the ice volume expressed in meters of sea level equivalent and time $t$ is in years. This formula gives significant

increase in iron flux for the case when sea level drops below 50 m, that is likely related to the fact that Patagonian dust source is very sensitive to the area of exposed shelf and glacial erosion processes. Numerical parameters in this formula were obtained by fitting simulated $D$ to the dust concentration in Antarctic record. This allows us to use the same parameterization for the iron fertilization effect in one-way and fully interactive experiments. To prevent large fluctuations in the iron flux related to fluctuations of time derivative of $S$, the dust deposition $D$ computed by this equation has been smoothed by

applying relaxation procedure. Namely, at each time step $n$, the dust deposition $D_n$ is computed as

$$D_n = (1-\varepsilon)D + \varepsilon D_{n-1}$$

Where $\varepsilon=0.001$ which is approximately equivalent to introducing of 1000 years filter.

### *A2.2 Dependence of remineralization depth on temperature*

In CLIMBER-2 the vertical profile of carbon below the euphotic zone is given by the formula

$$f(z) = \left(\frac{z}{z_r}\right)^{0.858}$$

where remineralization depth $z_r$ is hold constant equal to 100 m. To take into account dependence of remineralization rate on ambient temperature, following (Segschneider and Bendtsen, 2013) we now use dependence of $z_r$ on the thermocline temperature (300m) $T$:

$$z_r = z_{ro}\ 2^{\frac{T-T_0}{10}},$$

where $T_0= 9^{\mathrm{o}}$C and $z_{ro}$=100 m. The value of $T_0$ was selected such that introducing of temperature-dependent remineralization depth does not affect atmospheric $CO_2$ concentration under preindustrial climate conditions. During glacial times temperature in the thermocline decreases by 2-3$^{\mathrm{o}}$C which causes increase of $z_r$ by 20-30%. This results in additional $CO_2$ drawdown by ca. 15 ppm.

### *A2.3 Parameterization of iron fertilization effect*
The rate of dust deposition which is prescribed from the ice cores in one-way coupled experiment or computed from global ice volume in fully interactive experiments is considered to be a proxy for iron flux and is used in parameterization of iron fertilization mechanism. This parameterization is only applied to the Southern Ocean (south of 40$^{\mathrm{o}}$S). As described in Brovkin et al. (2002), net primary production of phytoplankton $\Pi$ in the model is described as

$$\Pi = c(D)\,r(T,R)\frac{P}{P+P_0}C_p(1-f),$$

where $C_p$ is the phytoplankton concentration, $P$ is the phosphorus concentration in euphotic zone, $r$ is a function of temperature $T$ and photosynthetic active insolation $R$, $f$ is the fraction of grid cell covered by sea ice, $P_o$ is a constant and c is a function of normalized dust deposition rate $D$. Note that in the case of prescribed dust deposition rate, $D$ was obtained by

multiplying observed dust concentration in mg/g units by factor $10^{-3}$ . North of $40^{o}$S, parameter $c$ is set to 1 and south of $40^{o}$S

$$c = \min(1, 0.1 + c_d D) \ .$$

With $c_d$=2, during glacial maxima the value $c$ reaches one thta implies that at these periods there is no iron limitation in the Southern Ocean. During interglacials, when $D$ is much smaller than 100, $c$ is close 0.1. Parameters of this parameterization were selected to reproduce present-day nutrients concentration in the Southern Ocean, and to obtain about 20-30 ppm additional $CO_2$ drop during glacial maxima due to the iron fertilization effect.

## A3. Variable volcanic outgassing

Following the idea by Huybers and Langmuir (2009) which has been tested already in Roth and Joos (2012), we introduced
15  a dependence of volcanic $CO_2$ outgassing $O$ on the rate of sea level change. Namely, we assume that volcanic outgassing linearly depends on sea level derivative with the time delay of about 5000 years:

$$O(t) = O_1\left(1 - O_2 \frac{dS(t-5000)}{dt}\right).$$

Here $O_1$=5.3 Tmol C yr$^{-1}$, and $O_2$=50 Tmol C m$^{-1}$. With these parameters volcanic outgassing does not change by more than 30% during all glacial cycles. Note, that over one glacial cycle the average value of $O$ is very close to $O_1$.

## A4. Modifications of the energy and surface mass balance interface

In our previous simulations with CLIMBER-2 we found that if maximum ice sheet volume in the Northern Hemisphere
25  exceeds 100 m, the AMOC remains in the off mode during the entire deglaciation. Although it may be realistic for some recent deglaciations, such long AMOC shutdown prevents simulation of complete deglaciation of North America. This is related to the fact that due to a very coarse spatial resolution of CLIMBER-2 linear interpolation of surface temperature between neighbouring sectors (American and Atlantic) cause a strong cooling over eastern Part of Laurentide ice sheet doe to the AMOC shutdown (see for example Arz et al., 2007). In the high resolution climate models, the effect of AMOC
30  shutdown on North America is rather limited compare to Europe (e.g. Zhang et al., 2014; Swingedouw et al., 2009). This is explained predominantly eastward direction of air masses transport. To compensate this resolution-related problem we made

magnitude of temperature anomaly correction over eastern North America (see Fig. 2 in Ganopolski et al., 2010) dependent on the strength of the AMOC. Namely, for the AMOC strength below 10Sv, the amplitude of temperature correction is scaled down by factor $\Psi_{max}/10$, where $\Psi_{max}$ is the maximum of the meridional overturning stream function (in Sv, 1 Sv=$10^6$ m$^3$/s) in the Atlantic Ocean. With this parameterization, during complete shutdown of the AMOC cooling over eastern North America is compensated by reducing of temperature correction. Introducing of this procedure minimizes the impact of the AMOC on Laurentide ice sheets mass balance. As the result, even prolonged AMOC shutdown does not prevent complete melting of the Laurentide ice sheet during glacial terminations.

**Table 1**. Model experiments performed in this study. P denotes prescribed characteristic, I - interactive, STD - standard model configuration, RD - variable remineralization depth, VO – variable volcanic outgassing, IF - iron fertilization in the South Ocean, TC – terrestrial carbon cycle, BR - brine rejection mechanism. Minus sign means that the process is excluded and plus sign means that process is included. Ice sheets are interactive in all simulations.

| Experiment | Radiative forcing of GHGs | Southern Ocean dust | Atlantic freshwater factor | Modell configuration |
|---|---|---|---|---|
| *400,000 yr experiments* | | | | |
| ONE_1.0 | P | P | 1 | STD |
| ONE_1.1 | P | P | 1.1 | STD |
| ONE_S1 | P | P | 1.1 | STD-RD |
| ONE_S2 | P | P | 1.1 | STD-RD-VO |
| ONE_S3 | P | P | 1.1 | STD-RD-VO-IF |
| ONE_S4 | P | P | 1.1 | STD-RD-VO-IF-TC |
| ONE_240 | 240 ppm | P | 1 | STD |
| INTER_1.0 | I | I | 1 | STD |
| INTER_1.1 | I | I | 1.1 | STD |
| *130,000 yr experiments* | | | | |
| ONE_1.0_130K | P | P | 1 | STD |
| ONE_1.1__130K | P | P | 1.1 | STD |
| ONE_BRINE_130K | P | P | 1 | STD+BR |

**Table 2.** "The carbon stew" at the LGM and the entire 400 kyr period

|  | LGM (22-19 ka) ppm | 400-0 ka ppm |
|---|---|---|
| Physical process + carbonate chemistry[*] | 57 | 38 |
| Ocean volume change | -12 | -6 |
| Terrestrial carbon storage | -13 | -8 |
| Iron fertilization | 22 | 6 |
| Remineralization depth | 15 | 10 |
| Volcanic outgassing | 8 | 3 |
| Total | 77 | 43 |

[*] deep ocean and shallow water carbonate sediments, carbonate and silicate weathering

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

**Figures**

a)                                                    b)

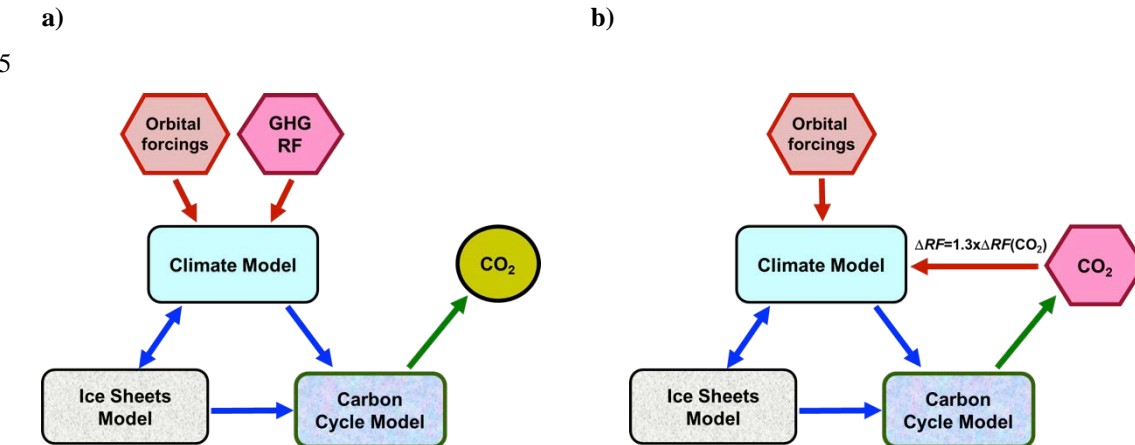

**Figure 1.** Coupling strategy. a) one-way coupled experiment; b) fully interactive experiment.

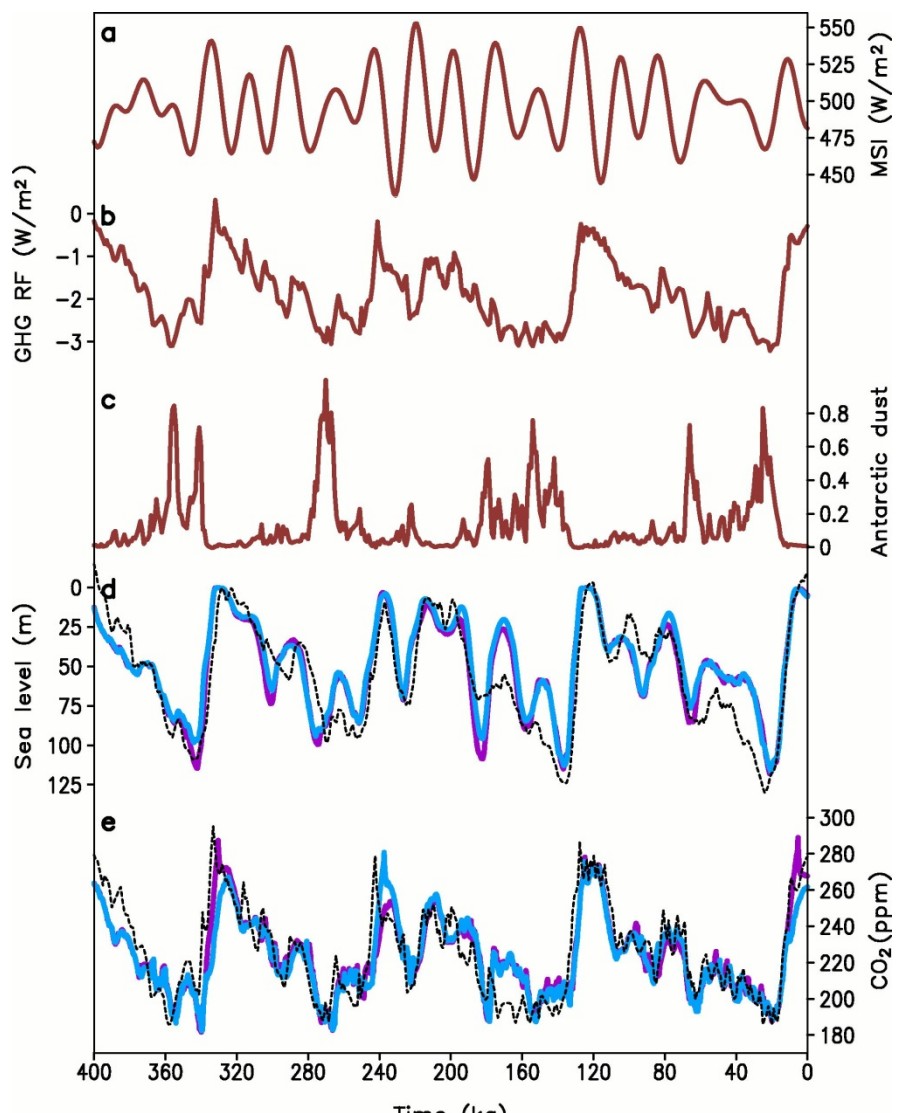

**Figure 2**. Transient simulations of the last four glacial cycles forced by orbital variations, observed concentration of well-mixed GHGs and dust deposition rate (one-way coupled experiments). a) Maximum summer insolation at 65°N, W/m²; b) radiative forcing (relative to preindustrial) of well-mixed GHGs, W/m²; c) Antarctic dust deposition rate in relative units; d) global ice volume expressed in sea level equivalent (m); e) atmospheric $CO_2$ concentration (ppm). Dark red colour in (a-c) represents prescribed forcings. Black dashed lines in (d) is sea level stack from Spratt and Lisiecki (2016), in (e) compiled Antarctic $CO_2$ record from Lüthi et al. (2008). Radiative forcing of GHGs in (b) is from Ganopolski and Calov (2011). Antarctic dust is from Augustin et al. (2004).  Blue lines in (d, e) correspond to the baseline experiment ONE_1.0 and pink lines to the experiment ONE_1.1 where meltwater flux into Atlantic was scaled up by factor 1.1.

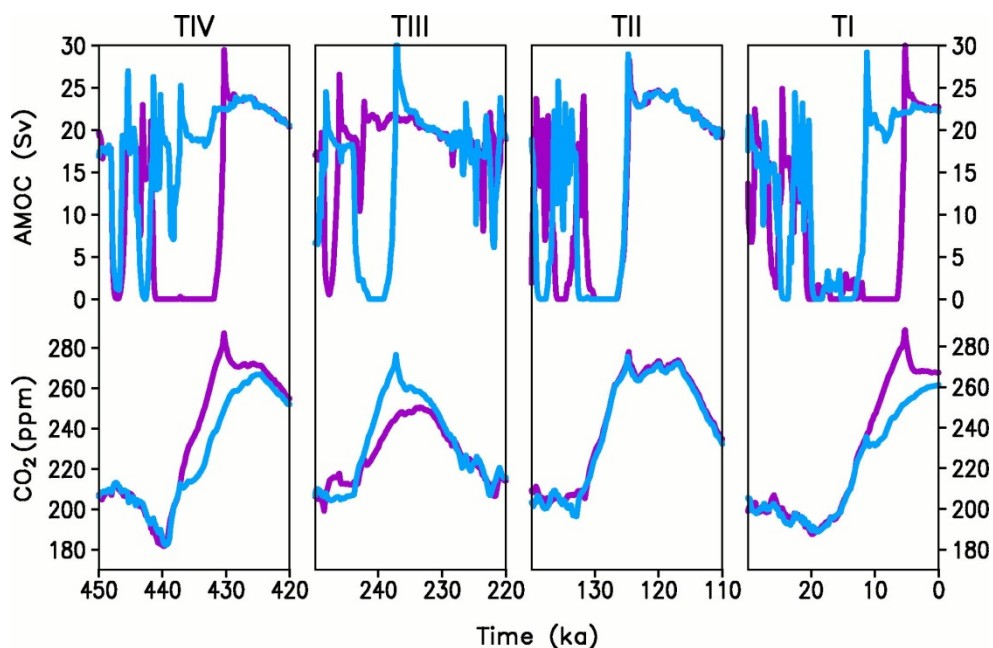

**Figure 3**. Temporal evolution of the AMOC, Sv (a), and atmospheric $CO_2$ concentration, ppm (b) during the last four glacial terminations. Blue lines correspond to the experiment ONE_1.0 and pink lines to the experiment ONE_1.1 where meltwater flux into Atlantic was scaled up by factor 1.1.

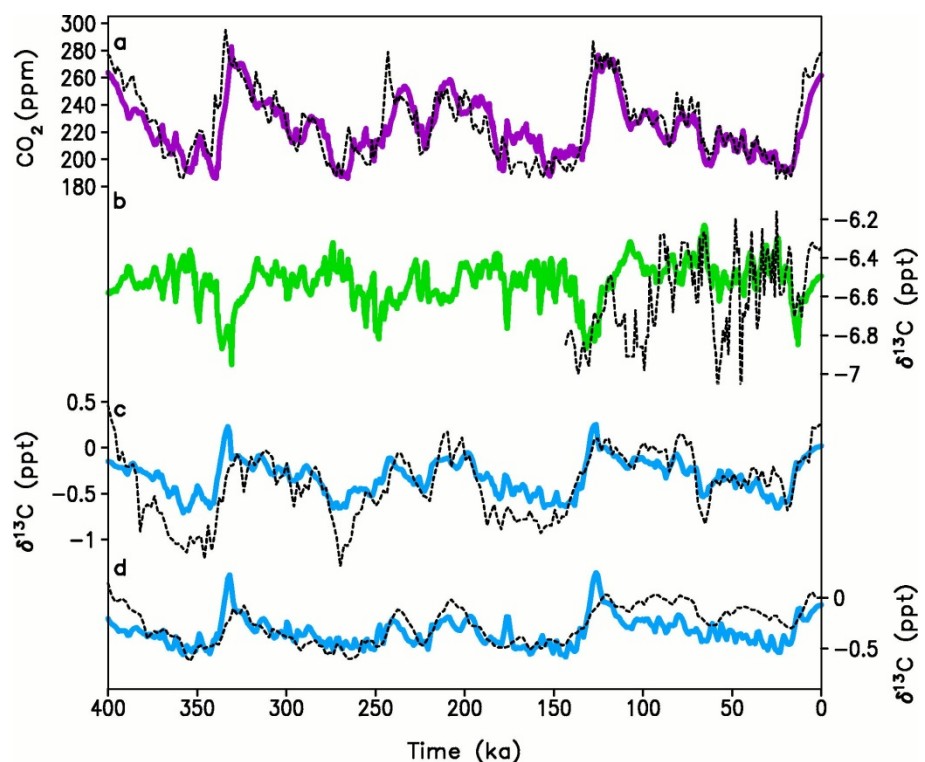

**Figure 4**. Simulated $CO_2$ and $\delta^{13}C$ with the one-way coupled model (ONE_1.0). a) $CO_2$ concentration (ppm) ref, b) atmospheric $\delta^{13}CO_2$ (‰), c) deep South Atlantic $\delta^{13}C$ (‰); d) deep North Pacific $\delta^{13}C$ (‰). Color lines – model results. Empirical data (black dashed lines): a) Lüthi et al. (2008); b) Egglestone et al. (2016); c) and d) Lisiecki et al. (2008).

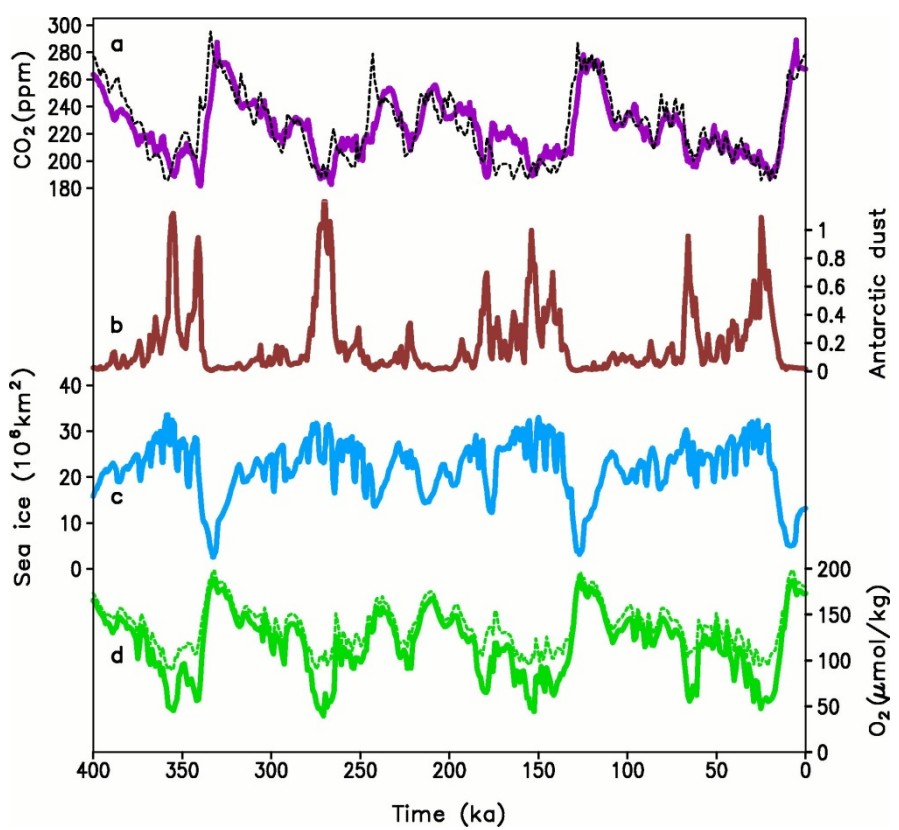

5    **Figure 5.** a) Simulated $CO_2$ concentration (ppm); b) prescribed Antarctic dust deposition rate in relative units; c) simulate annual mean sea ice area in the Southern Hemisphere ($10^6$ km$^2$); d) simulated oxygen concentration in the deep South Ocean in (μmol/kg) in the ONE_1.1 experiment (solid line)  and the identical experiment but without iron fertilization effect (dashed line).

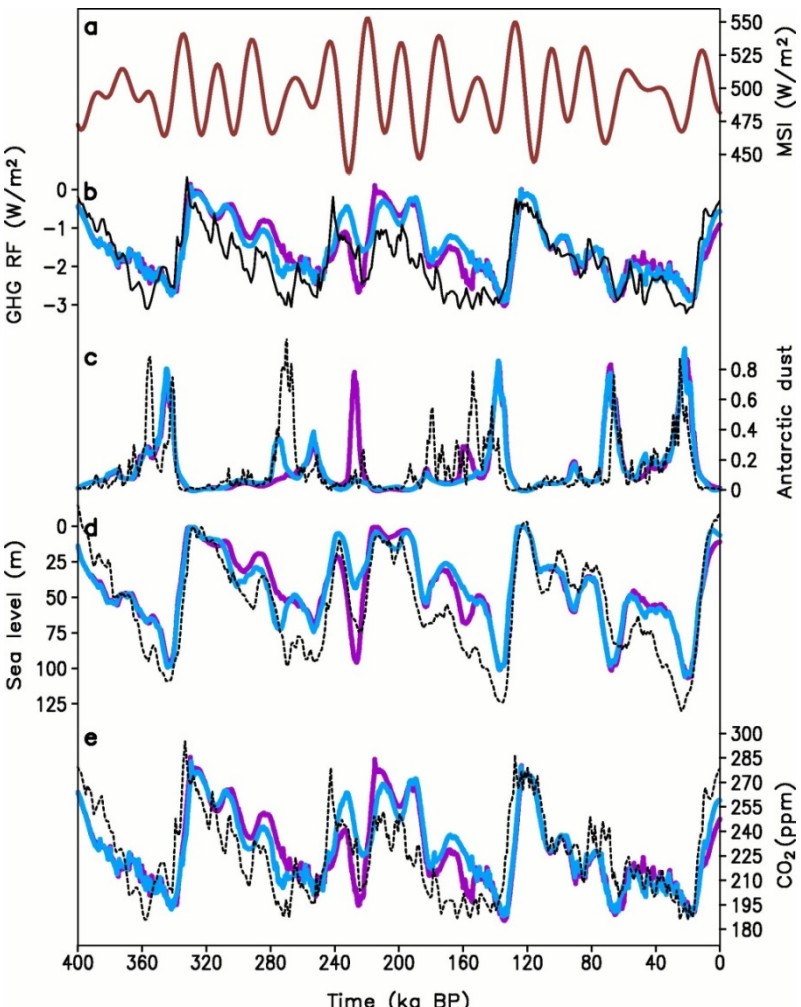

**Figure 6**. Transient simulations of the last four glacial cycles forced by orbital variations only (fully interactive experiments) a) Maximum summer insolation at $65^o$N, W/m$^2$; b) radiative forcing (relative to preindustrial) of well-mixed GHGs, W/m$^2$; c) Antarctic dust deposition rate in relative units; d) global ice volume expressed in sea level equivalent, m; e) atmospheric $CO_2$ concentration, ppm. Black line in (b) is radiative forcing of GHGs from Ganopolski and Calov (2011). Black dashed lines in (c) is Antarctic dust is from (Augustin et al., 2004), in (d) is sea level stack from Spratt and Lisiecki (2016), in (e) compiled Antarctic $CO_2$ record from Lüthi et al. (2008). Blue lines in (d, e) correspond to the fully interactive experiment INTER_1. and pink lines to the experiment INTER_1.1 where meltwater flux into Atlantic was scaled up by factor 1.1.

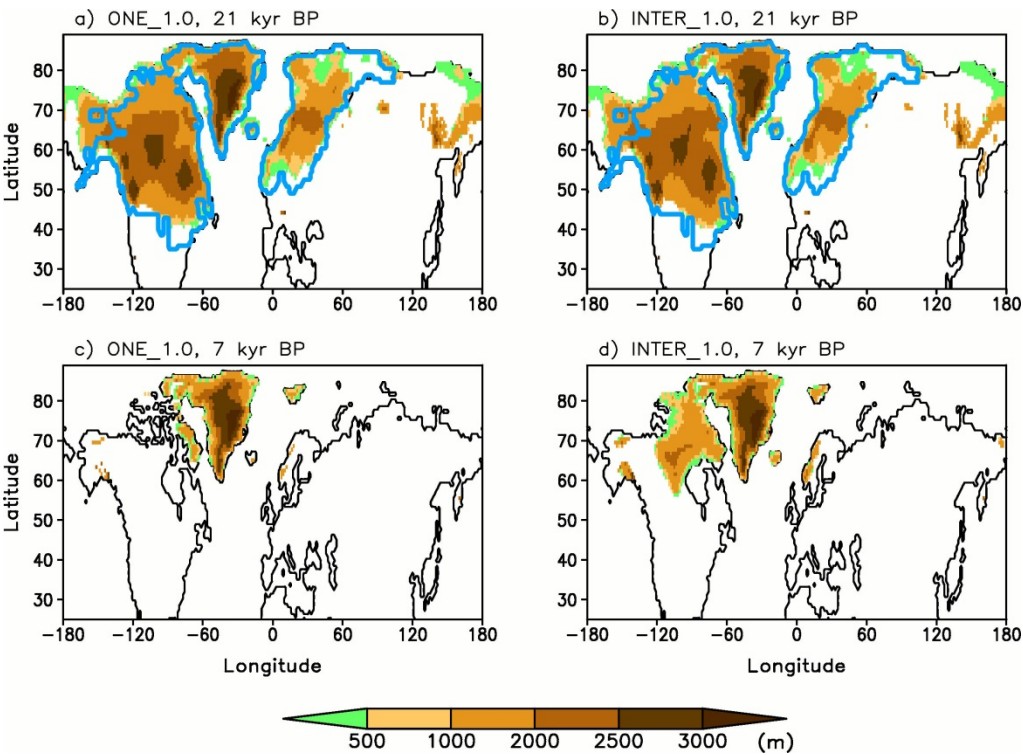

**Figure 7**. Simulated ice sheets elevation, m, at 21 ka (a, b) and 7 ka (c,d) in the one-way coupled experiment ONE-1.0 (a, c) and fully interactive experiment INTER-1.0 (b, d). Blue lines represent Ice-5g reconstruction at the LGM (Peltier, 2004).

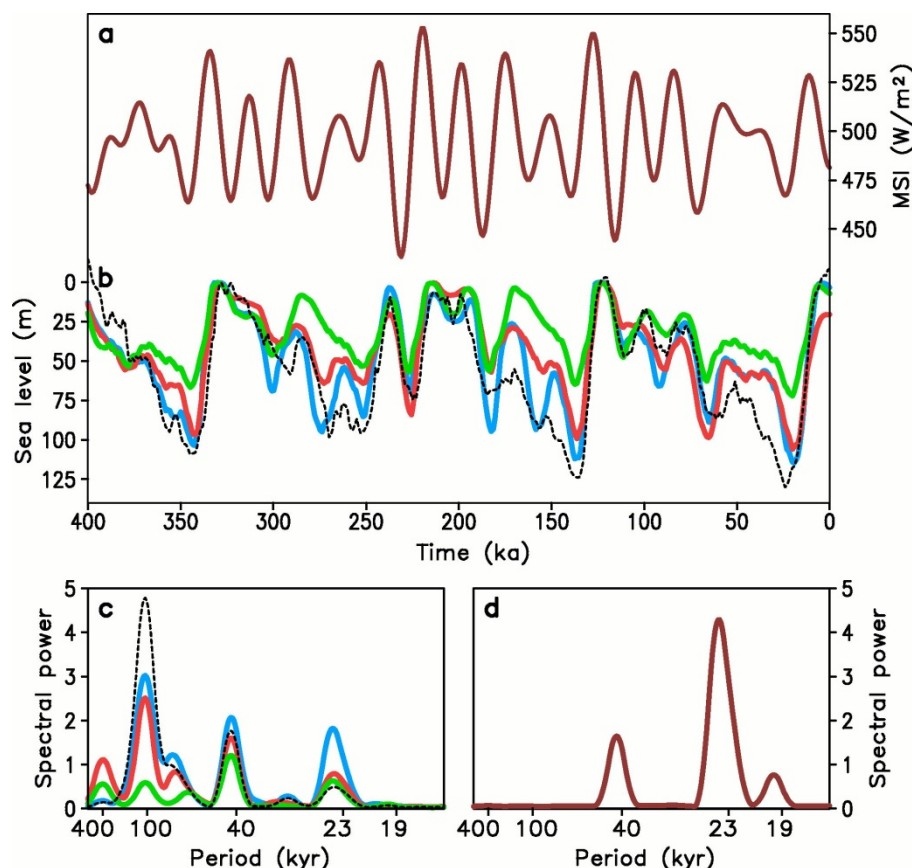

**Figure 8**. Transient simulations of the last four glacial cycles forced by orbital variations, with prescribed, interactive and fixed concentrations of well-mixed GHGs. a) Maximum summer insolation at 65°N, W/m²; (b) Temporal evolution of reconstructed and simulated sea level, m; (c) frequency spectra of the global ice volume; (d) frequency spectra of boreal summer insolation. Black line is for the data (Spratt and Lisiecki , 2016), blue line corresponds to the one-way coupled experiment ONE_1.0, red line to the fully interactive experiment INTER_1.0, and green line to the ONE_240 experiment with constant (240 ppm) $CO_2$ concentration; d) frequency spectra of orbital forcing.

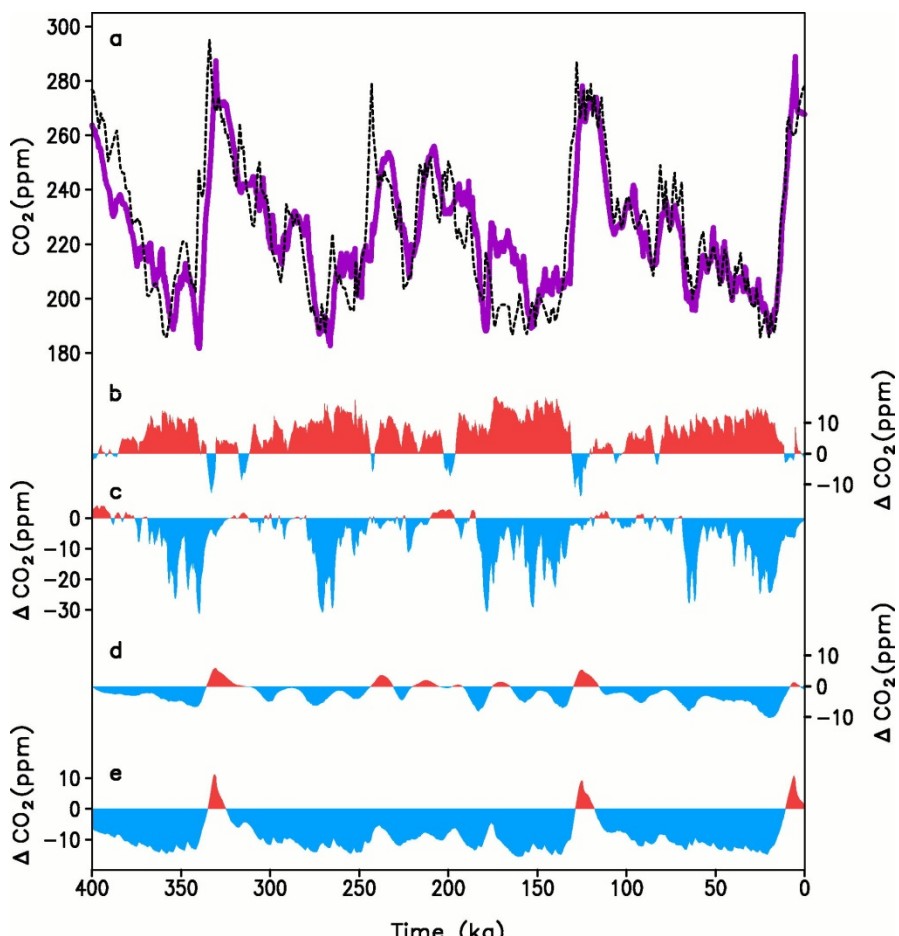

Figure 9. Results of factor separation analysis. a) Simulated $CO_2$ (ppm) in one-way coupled ONE_1.1 experiment (purple line) and reconstructed $CO_2$ concentrations (black dashed line, Lüthi et al., 2008). b-d): contributions to simulated atmospheric $CO_2$ (ppm) of terrestrial carbon cycle (b), ONE_S4 – ONE_S3; iron fertilization (c), ONE_S3 – ONE_S2; variable volcanic outgassing (d), ONE_S2 – ONE_S1; temperature-dependent remineralization depth (e), ONE_S1 – ONE_1.1.

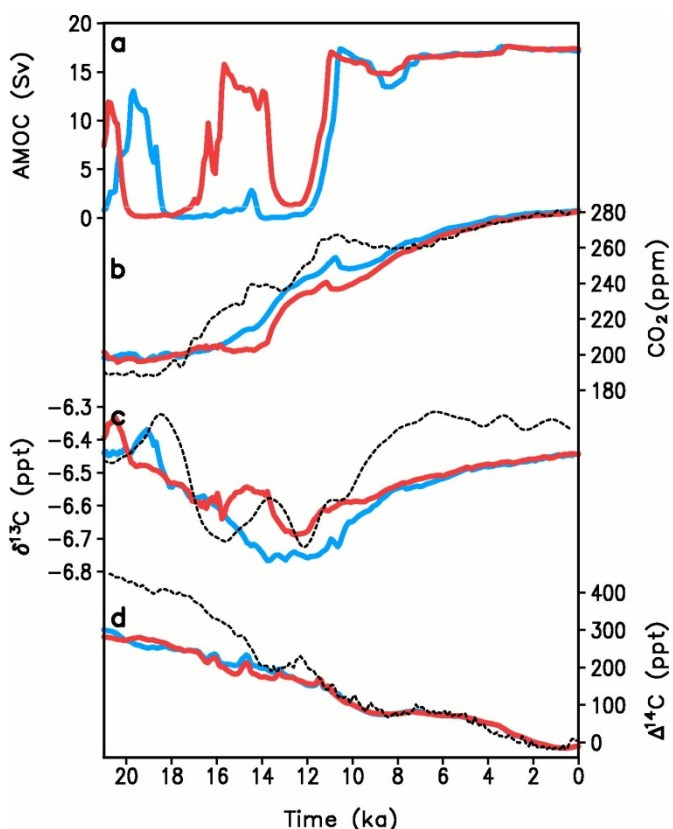

5 **Figure 10**. Simulation of Termination I with the set of one-way coupled models which differs only by scaling of freshwater flux. Blue line corresponds to the ONE_1.1_130K experiment with scaling factor 1.1, red line – the ONE_1.0_130K experiment with scaling factor 1.0. a) AMOC strength, Sv; b) atmospheric $CO_2$, ppm; c) atmospheric $\delta^{13}CO_2$, ‰; d) atmospheric $\Delta^{14}CO_2$, ‰. Dashed lines: b-c) ice-core data (Lüthi et al., 2008; Schmitt et al, 2012); d) IntCal13 radiocarbon calibration curve (Reimer et al. 2013).

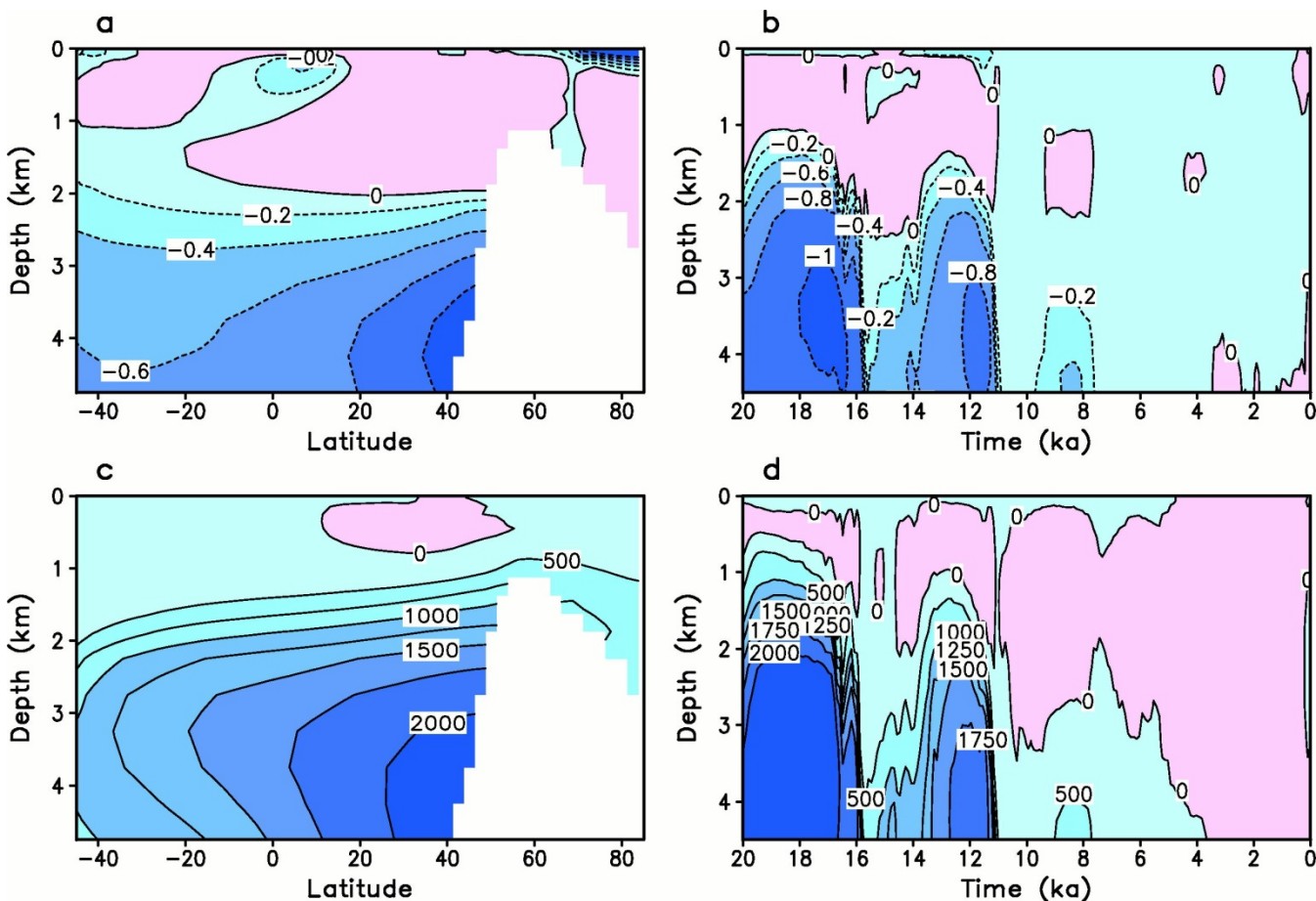

**Figure 11**. $\delta^{13}C$ and radiocarbon ventilation age  distribution in the Atlantic ocean in the ONE_1.0_130K simulation of Termination I. (a, b) $\delta^{13}C$, ‰, (c, d) radiocarbon ventilation age  in yr $^{14}C$. (a, c) differences between LGM (21 ka) and pre-industrial in the Atlantic ocean. (b, d) temporal evolution of anomalies during the past 20 ka at 20°N in Atlantic.

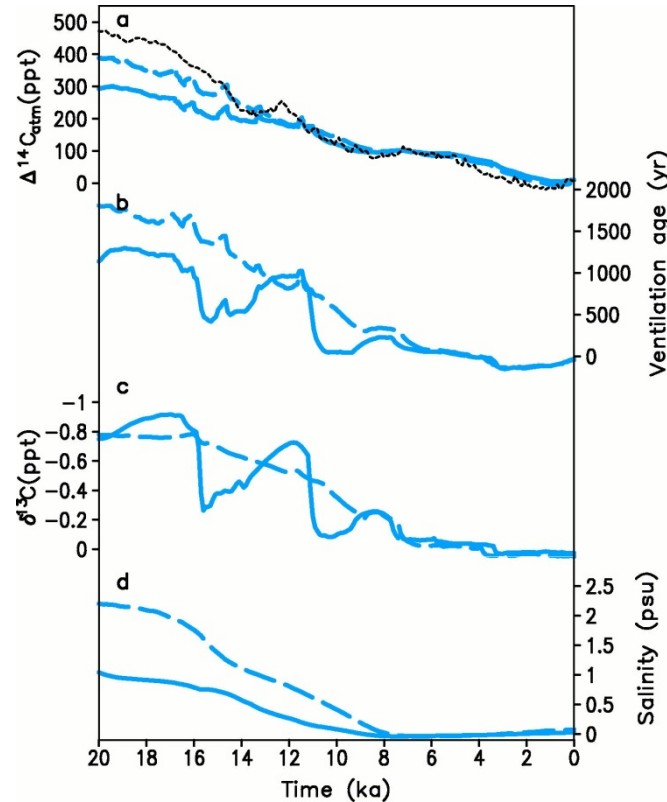

**Figure 12**. Simulation of Termination I in the standard ONE_1.0_130K experiment (solid blue) and the ONE_BRINE_130K (dashed blue) experiment which includes brine parameterization and stratification-dependent vertical mixing (g). (a) atmospheric $\Delta^{14}$C (in ‰). (b) Deep tropical Atlantic $\Delta\Delta^{14}$C, ‰. (c) Deep tropical Atlantic $\delta^{13}$C, ‰. (d) Deep Southern Ocean salinity, psu. (c-d) are for the depth 4 km. Black dashed line is IntCal13 radiocarbon calibration curve (Reimer et al.

10    2013).

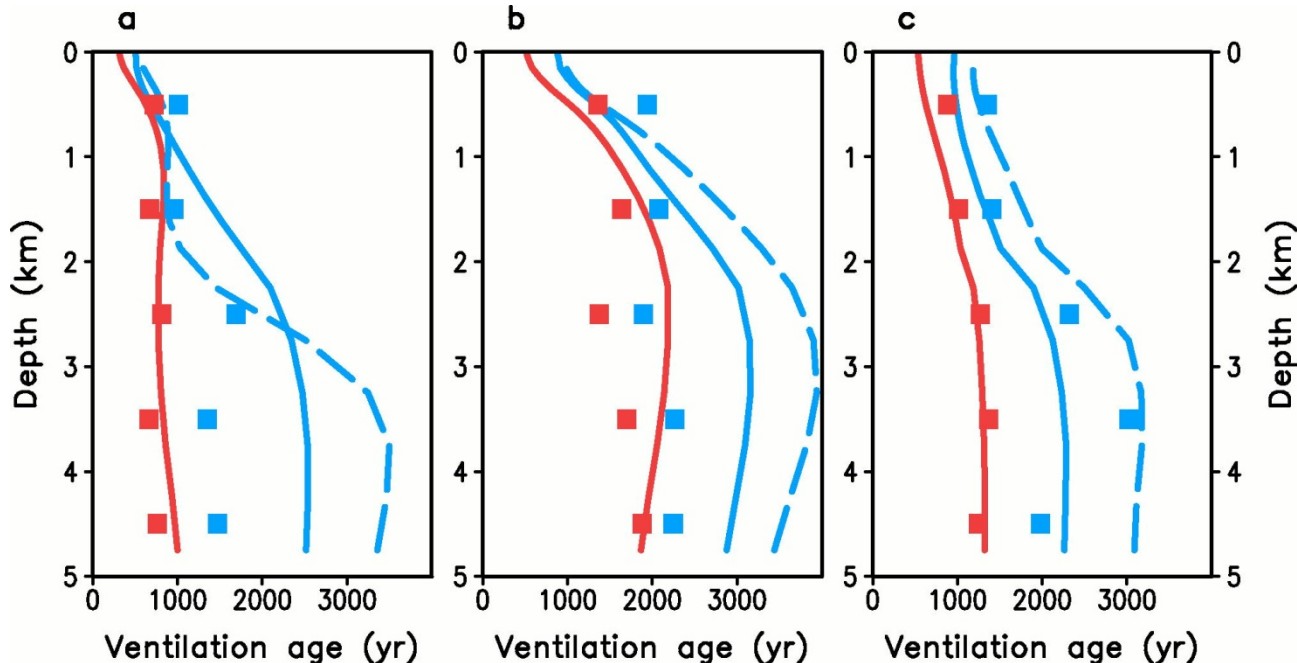

**Figure 13**. Vertical profile of ventilation age in $^{14}$C years for Atlantic (a), Pacific (b) and Southern Ocean (c). Red line represents modern conditions, solid blue – LGM in ONE_1.0_130K experiment using the standard version of the model, dashed blue – LGM in ONE_BRINE_130K experiment with the model version which includes brine parameterization and stratification-dependent vertical mixing. Red (blue) squares represent basin averaged radiocarbon age for modern (LGM) state based on the data from Skinner et al. (2017).

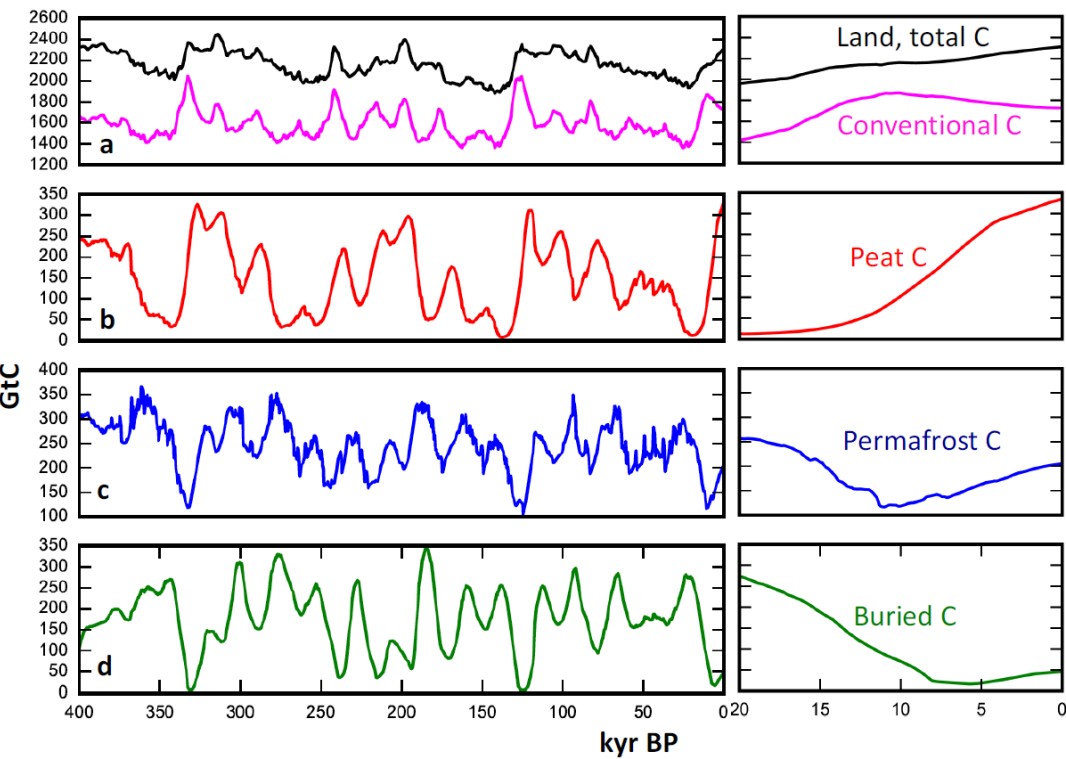

**Figure 14**. Dynamics of terrestrial carbon pools (Gt C) in the one-way coupled ONE_1.0 simulation. Left, the whole 400 kyr period; right, the Termination I period. a) Black line – total carbon storage; magenta line - conventional carbon pools (biomass and mineral soils), (b-d): peat, permafrost, and buried carbon storages, respectively.