# Peer review of "Simulation of climate, ice sheets and CO2 evolution during the last four glacial cycles with an Earth system model of intermediate complexity"

_Climate of the Past, 2017_

## Referee Comment (RC1) · Anonymous Referee #1 · 17 May 2017

Ganopolski and Brovkin simulate four Glacial/interglacial (G/IG) cycles with the model of intermediate complexity CLIMBER2 in both a fully interactive and 1 way coupled mode. In both set ups, the model is able to reproduce the major features of G/IG cycles: i.e. changes in sea-level, ice-sheet extent (and volume), atmospheric CO2... It is an interesting study, certainly worth publishing in Climate of the Past. My main comment would be that I don't find the goals or conclusions of the study very clear. The manuscript tries to tackle various aspects of G/IG cycles but without going deeply in any of them. The authors are rightly very careful in not over-interpreting or making hasted conclusions from their results because the model used is quite simple. But

maybe the study would gain in visibility by focusing on fewer points and putting them in a broader context. Please find additional comments below.

1) The first part of the introduction suggests that the radiative role of CO2 in driving 100kyrs G/IG cycles is controversial. To explore this, a simulation with constant pCO2 (240ppm) is performed. It leads to G/IG variations with ~50% full G/IG amplitude and with dominant periodicity of 40ka. To me, this would tend to highlight the dominant role of CO2 in driving 100ka cycles, but this result or its implications are not really discussed.

2) CO2 changes:

The study simulates full G/IG changes in pCO2 due to a combination of processes and in global agreement with previous studies. However, due to the relative simplicity of the model and its configuration (zonally-averaged basin), I would think that the impact on pCO2 of oceanic circulation changes, sea-ice and wind related changes are under-estimated, while iron fertilization changes are overestimated. In addition, I am a bit surprised not to see any mention of the impact of changes in the carbonate system (e.g. shallow water carbonate deposition). A few studies (see A. Ridgwell or F. Joos studies) have shown that this has a significant impact on pCO2 particularly at the end of the deglaciations (early interglacial) and thus also glacial inceptions.

The authors highlight the impact of deglacial AMOC changes on the shape of the pCO2 trajectory at the end of the deglacial phase. This is an interesting aspect but: i) Its reasons are not discussed in details ii) Can we really believe it given that the shape and timing of the deglacial CO2 changes are not represented correctly and some processes are likely missing or misrepresented (e.g. shallow water carbonate deposition, oceanic circulation changes).

3) Minor Changes in weathering and its impact on pCO2 are not very clear. I realize it is mentioned in Brovkin et al., 2012, but maybe a brief description might be useful.

Figure legends: Please make sure all appropriate references for the proxy are included in figure legends. For example Antarctic dust in Figure 2, Figure 4c... Proxy for atm. d13CO2 could be included in Figure 4b, even if they only cover part of the last G/IG cycle. Figure 8a: purple line.

———————————————————

---

## Referee Comment (RC2) · L. Skinner (Referee) · 26 Jun 2017

This modelling study is truly fascinating and contains a wealth of hypotheses and information to explore. This represents a major strength of what is a very interesting paper of course, but it also presents something of a challenge, not least with respect to maintaining complete clarity on all possible points that readers might wish to see explored and explained. I believe this to be a valuable and informative contribution that should eventually be published in CP; however, I provide some comments below, which I hope may prove helpful in revising the manuscript prior to publication.

[Figure]

My most general comment is that that study appears to focus overly on the 'success' of the numerical model simulations (and therefore the apparent success of the many model \*choices\* that have been implemented), rather than the justification or otherwise of the choices that have been made, for example as attested to by proxy data. In other words, it may well be that a viable 'recipe' for glacial-interglacial CO2 has been devised, but how do we know it is the right one? Arguably the only way to explore the latter question is to compare the biogeochemical/physical 'fingerprints' of that recipe with proxy data. My feeling is that more could (and probably should) be done in this regard, in particular with respect to carbonate chemistry, radiocarbon, oxygen and nutrient distributions/trajectories. Indeed, I would suggest that even if proxy data are too sparse to comprehensively test the particular 'CO2 recipe' that is adopted in this study (or if it is too much work to compile the data needed for this, since arguably this could be beyond the scope of this initial study), it should still be possible to identify its 'biogeochemical fingerprints' so that eventually the recipe we are being offered can be tested by others. Without this we are left without the means of assessing whether or not the CO2 recipe in this study is not only viable, but also possibly correct. I would propose that three specific parameters to possibly consider in more detail are: radiocarbon, carbonate chemistry and oxygenation/respired carbon. Of these, radiocarbon and carbonate chemistry offer the best opportunities for data-model comparisons. I return to these suggestions below.

Specific comments:

1. The abstract states that the co-evolution of climate, ice-sheets, and carbon cycle have been simulated over 400,000 years using insolation as the only external forcing. This is an impressive feat, and the reader wonders how this has been achieved of course; what are the key processes and feedback loops at the heart of the long-standing 'mystery of the ice ages'? It would be helpful if the abstract summarised the authors proposal succinctly. More specifically, it seems that a successful simulation of climate, ice volume and atmospheric CO2 has been achieved by appropriately

scaling the rate of change of atmospheric CO2 to ice volume (using parameteriza-tions for iron fertilisation and volcanic CO2 outgassing), and by further implementing additional climate-carbon cycle feedbacks that operate primarily through temperature-dependent respiration rates in the ocean, marine CO2 solubility effects and ocean circulation changes. The extent to which the phenomena have a been implemented as modelling choices, and the extent to which the magnitude of their impacts (e.g. on CO2) depends on parameter choices, should be made clear.

2. The abstract focuses on the deglaciation as being particularly sensitive to parameter choices, apparently in contrast to the rest of the glacial cycle (for which many features are argued to be 'rather robust'). I feel that this might be a little misleading; can the meaning of this statement be clarified? In what sense exactly can modelled features be said to be robust?

3. The issue of CO2 overshoot: this is highlighted in the abstract as a key finding, but it needs to be explained more fully I think. Why exactly does this phenomenon occur? Does it depend on model choices and if so which ones, or is it a fundamental aspect of the physics in the model? It would appear that the AMOC is sensitive to freshwater forcing throughout the deglaciation, but that AMOC anomalies early in the deglaciation (and during the glacial?) have no appreciable carbon cycle impact; why is this? Is marine soft tissue pump efficiency 'maxed out' (exhausted) and therefore insensitive to further enhancement until the parameterizations for increased Fe-fertilisation and nu-trient respiration rate are released? More explanation is needed for this phenomenon, especially if it is highlighted as being particularly noteworthy.

4. Page 4, Line 11: the way in which iron fertilisation is implemented needs to be clarified. How is exactly is nutrient utilisation scaled with dust and on what basis? How do we know that the right scaling has been applied, or is it essentially arbitrary? Can the scaling be justified on the basis of nitrogen isotopes (simulated) or anything else? Without such details the iron fertilisation mechanism will always seem like a sort of 'magic bullet' for drawing down carbon into the ocean.

5. Page 4, Line 23: the implementation of radiocarbon in the model should be explained a little more clearly too (e.g. is it simulated as an isotope tracer that undergoes gas exchange, fractionation etc... or is it a pseudo-tracer with a decay timescale that is restored to a particular value at the ocean surface?). Note that Hain et al. (2014) did not produce a radiocarbon production scenario; please check this reference (ultimately the production scenario will be based on Be-10 or geomagnetic field strength and the original references should be cited). In general I think that more should be made of the radiocarbon outputs, e.g. in comparison with existing data. Such data should be added to figure 12 for example, and any agreement/disagreement discussed. I return to this later.

6. Page 5, Line 5: notably this way of doing greenhouse gases will produce incorrect results for millennial timescales, since methane and CO2 are not in phase during D-O/Heinrich events. Does this matter; can it be shown that it does not matter?

7. Page 5, Line 22: can this careful calibration of volcanic outgassing be tested against e.g. atmospheric d13CO2 for example (note that these data are available for the last glacial cycle from Eggleston et al. 2016)? If the volcanic control on atmospheric CO2 is so strong, it might also be expected to affect the isotopic composition of the atmosphere quite strongly (as well as the deep ocean carbonate system - more on this later). Is the surface (i.e. non-solid Earth) carbon cycle balanced; i.e. is 5.3TmolC/yr going back into the solid earth in the model? All of these are important questions that jump out at the reader, but are not dealt with at all in the current manuscript.

8. Page 5, Line 30: this procedure for 'initial condition conditioning' is very interesting, but it is not so obvious why the system should converge on the same initial and final states, regardless of the history of evolving boundary conditions over 410kyrs; is it possible to clarify? What component is drifting that depends on the state of the system (and that eventually reaches an equilibrium through this iterative process)?

9. Page 6, Line 30: the lag of CO2 is an important clue as to what is (perhaps) not

right in the model parameterizations that have been selected. One wonders if this has something to do with the choice to scale iron fertilisation with ice volume: dust does not track sea level very closely in reality, and more specifically it drops off rapidly before sea level has risen much in the deglaciation. Or is the lag due to something else? More analysis of the source of this mismatch would be illuminating (more illuminating than if the model happened, perhaps accidentally, to match observations perfectly).

10. Page 6, Line 32: as noted above, a more detailed explanation is necessary for the mechanisms underlying the overshoot in CO2, and for CO2 release as a function of AMOC variability in general. There is not universal agreement amongst models for millennial scale controls on atmospheric CO2 and the role of the AMOC, so it will be useful to know what is going on in this particular model experiment, and why the carbon cycle response to AMOC changes is so context dependent. On page 9 it is suggested that the CO2 overshoots depend primarily on remineralisation depth changes that in turn stem from subsurface heat anomalies, but this is not clearly stated or explored anywhere else.

11. Page 7, Line 20: again, this lag, and it's increase in the fully coupled runs is important, and should be diagnosed more clearly, as it is telling us something important about the model choices that have been implemented.

12. Page 8, Line 4: it is very interesting and important that CO2 changes on a dominantly 100ka timescale are not needed to produce glacial cycle sin the model, but where does the 100kyr timescale for ice sheet growth/decay come from in this model; is it simply the timescale at which the ice sheets get big enough for the dirty-ice albedo instability to kick in? If so, how is that feedback constrained (is the time scale a model choice once again or is it due to a fundamental limitation on ice growth rates and basal sliding etc...); how do we know it should happen on that timescale?

13. Page 8, Line 24: can it be stated that the 'better' performance of the enhanced freshwater flux experiments indicates an under-representation (or misrepresentation)

of the role of ocean circulation perturbations, at glacial transitions in particular? Is it possible that it could also be that this enhanced forcing is needed to compensate for other biases, e.g. from iron fertilisation or volcanic CO2 parameterizations? How would we know, what do we learn from this?

14. Page 9, Line 1-7: it would be helpful to include a table that clarifies the 'carbon stew' and the contribution of each mechanism that is implemented, e.g. based on average glacial and interglacial values.

15. Page 9, Line 14: the original references of Matsumoto (2007) and Matsumoto et al. (2007) are missing here, and where the notion of temperature dependent respiration rates is introduced.

16. Page 9, Line 25: some more detail on the volcanic CO2 implementation is needed; what about the balance of marine versus sub-aerial volcanism, and their different responses to ice vs water loading; how is this treated and on what basis is a particular magnitude of volcanic CO2 flux chosen? More justification/testing of the volcanic CO2 implementation is also needed; what is the impact on marine carbonate chemistry and does this tally with proxy evidence (it should cause marine carbonate ion concentrations to go up in the glacial, at odds with data from the Atlantic where it goes down, and the Pacific where it stays pretty constant)? Is there a longer-term feedback via carbonate preservation; are changes in volcanism perfectly balanced by weathering and sedimentary carbon outputs in the model, and if not what is compensating for the drift in global 'surface' carbon inventories that would result from this? Also, as noted above, please state what the impacts of the changing volcanic carbon fluxes on atmospheric carbon isotopes are: are they essentially nil?

17. Page 10, Line 11: it is stated that the brine rejection parameterization cannot be tested with observational data, but is this entirely true/fair, especially given the lack of testing offered in this study for the volcanism and temperature dependent respiration rate mechanisms? A critical analysis of all key modelling choices should be provided;

not just for brine rejection.

18. Page 11, on deglacial d13Catm: The text gives the impression that the deglacial d13Catm tends are quite accurately reproduced, but the match is not great. The 'W' in deglacial d13Catm is not particularly clear; what is this mismatch attributable to? Does it mean that the model is not simulating the correct marine carbon cycle response to AMOC change? Also, why is the more substantial early Holocene d13Catm rise seen in available data not reproduced; does this mean that terrestrial carbon uptake is too small in the model? Do marine carbonate ion values confirm this latter possibility or not (or at least demonstrate that marine carbonate ion reconstructions could be used to test the model)? I think a great deal more should be made of the isotope simulations and their comparison with proxy data.

19. Page 12, on deglacial 14C: Even more so than for the stable carbon isotopes, I think that a great deal more should be done with the radiocarbon simulations and their comparison with observations. Figure 10 should really include data, as should Figure 12 (this could be made substantially easier to include by a recent compilation by Skinner et al., Nature Communications, 2017). Radiocarbon data provide very strong constraints on the ocean state; if the simulation does not fit the available data, some discussion is warranted. This relates to the following section, where it emerges that the model simulation not preferred by authors, using brine rejection as a stratification mechanism, produces radiocarbon data that better fit the data (though again, no direct comparison with data is shown).

20. Page 12 , Line 14: it is stated that the radiocarbon data are in good agreement with Roberts et al. (2016); however that publication did not present radiocarbon data. Please correct the reference and/or clarify.

21. Page 12, Line 28: if the preferred model simulation does not fit the radiocarbon observations, does this not mean that the "CO2 stew" proposed in the manuscript must not be completely accurate? Please clarify.

22. Page 12, Line 30: in the manuscript DD14C is used as the preferred ventilation metric; however, this metric does not scale with the isotopic disequilibrium between two reservoir in a constant manner. In other words, a given DD14C value will reflect a different degree of isotopic disequilibrium (or ventilation age) depending on the absolute D14C. This not only makes DD14C a particularly confusing metric, but it also means that simulated DD14C values can match observed values without being correct if the absolute atmospheric/marine D14C values are too high/low. This indeed seems to be the case here, as the simulated D14Catm at the LGM is ∼150 permil lower than observed. For these reasons I would urge the authors to use marine vs atmosphere radiocarbon age offsets (B-Atm), which can also be converted to a ratio of isotopic ratios (or F14b-atm, Soulet et al., 2016) if a semblance of 'geochemicalness' is required.

23. Page 13, Section 5.3: can the authors state clearly what the implications are, if there are any, for marine and atmospheric carbon isotopes (13C, 14C) of the terrestrial carbon shifts, e.g. at the last deglaciation? It has been proposed that parts of the observed deglacial 14Catm record might be explained by permafrost changes; do the model results support a significant impact on deglacial atmospheric radiocarbon (or d13C)?

24. Page 13, Line 33: "..the model simulates the correct timing of glacial terminations..." I would suggest to be more precise (e.g. ice volume, but not CO2?), and perhaps to quantify this as being within a certain (millennial?) margin of error.

25. Page 14, Line 2: "...ocean carbon isotopes evolution is in agreement with empirical data." Should stable carbon isotopes be specified; should the statement be qualified somewhat (e.g. global spatial patterns have not been matched.. and the fit is assessed only in very general terms)?

26. Page 14, Line 3: should this read "the magnitude of atmospheric 14C change is underestimated"? And on Line 5, I would say that the statement regarding disagreement with data has not really been backed up very strongly as there is no illustration of

a comparison with data in the manuscript.

27. Page 14, Line 10: I think that some more explanation is required for what is meant by 'robust' in this context.

28. Page 16, Line 25: as noted above, the scaling of iron flux with sea-level is arguably questionable, since although dust fluxes in Antarctica increase relatively late, when sea level has fallen and CO2 has already dropped somewhat, it is also true that dust fluxes drop off very quickly on the deglaciation, before sea level as risen appreciably. Does this not mean that the 50m RSL threshold for dust changes is somewhat incorrect (i.e. it has the effect of keeping iron fertilisation strong for too late in the deglaciation)? A plot of how the timing of dust/iron fluxes in the model compare with the timing of dust fluxes in Antarctic ice cores might provide a test of this. I would suggestion including such a figure as a justification of the chosen parameterization. Again, I think that a clear description is needed for how export production is scaled to dust fluxes in the model, and on what basis the chosen scaling is justified (it would be nice to know what the Southern Ocean and global export productivity is in the model on average for glacial and interglacial states). How is iron release from dust simulated, how is biological activity as a function of iron availability simulated etc..? I think that a clear description of how biological carbon fixation/export is linked to dust fluxes should be included in the appendix.

29. Figure 4: atmospheric d13C data for the last glacial cycle and deglaciation should be added, including e.g. Eggleston et al. (Palaeoceanography, 2016).

30. Figure 7b: perhaps add the power spectrum for a appropriate insolation record, as a dashed line?

31. Figure 8: I personally would find it useful if the plots b-e were drawn as filled curves, either side of the zero line, so that it was clear when each process was acting as a source or sink for CO2.

[Figure]

32. Figure 9: I think this figure would benefit from adding a comparison between simulated and observed marine radiocarbon ventilation ages some key locations/regions. It may provide insights into why the atmospheric simulations do not match the observations.

33. Figure 10: why do the plots only go to 40oS? I think this figure would greatly benefit from added data comparison. For this it would be essential to convert the radiocarbon activities to radiocarbon age offsets or radiocarbon ratios (i.e. not relative deviation offsets).

34. Figure 11: probably it would be good to add an indication of what the green line is (even though it is obvious by process of elimination). Does the brine rejection experiment not include freshwater pulses during deglaciation; why does it not exhibit any deglacial anomalies at all? Again, data might usefully be added to the figure for comparison.

35. Figure 12: this figure is the most obvious one in which to include a comparison with observations, along with an addition to the text of a discussion of any mismatches between the various experiment outputs and the observations. It seems to me that if the simulation does not fit the data, then something is amiss, which we might learn from if it was identified.

36. Figure A1: What are the different coloured substrates? Perhaps more can be done with this figure?

I hope the above comments are useful to the authors; their quantity is testimony to how interesting this study is. I add below some purely editorial/grammatical suggestions that I hope the authors may also find useful:

1. Page 3, line 30: The ice sheet model is only applied...

2. Page 4, line 3: As shown in Ganopolski and Roche (2009), temporal dynamics.... in CLIMBER-2 are very sensitive... of freshwater flux to the North Atlantic.

3. Page 5, line 16: ...multiple glacial cycles represents a challenge..

4. Page 7, line 16: In particular, in the fully...

5. Page 8, line 11: ..faster ice growth during the initial part... very sensitive to ice volume.

6. Page 9, line 1: ...in our simulations they counteract glacial $CO_2$ drawdown... since terrestrial carbon contains ca. 350 Gt less carbon at the LGM...

7. Page 9, line 16: I would cite Matsumoto (2007) here...

8. Page 10, line 15: ..which is at odds with reality. This means that to be an efficient mechanism for ...at least by an order of magnitude.

9. Page 10, line 20: is there a paper by Miller et al. that would be appropriate here?

10. Page 11, line 8: note the need for Danish letters in Bolling-Allerod.

11. Page 11, line 18: ...maximum at the end of the North Atlantic cold event, which... ?

12. Page 11, line 22: ...at the beginning of the interglacial followed by... At the same time an earlier AMOC recovery causes only a temporary/brief... ?

13. Page 12, line 6: ...An alternative hypothesis..

14. Page 12, line 33: I would propose to cite more than just Freeman et al. (2016), since these authors only presented new data from the low-latitude and Northeast Atlantic.

15. Page 13, line 7: ..very old (likely to be at odds with palaeoclimate data)...

16. Page 14, line 2: ...ocean stable carbon isotopes...

17. Page 14, line 12: ..decreases the amplitude of glacial-interglacial...

---

## Author Comment (AC1) · 4 Aug 2017

**General response**

We thank both referees for useful and constructive comments and suggestions. Below is our response and original Referees comments in italic. Hereafter we will refer to our manuscript as GB17.

Both referees raised a number of questions primarily related to our choice of the "carbon stew". Reviewe#1 explicitly stated that he/she believe that influence of physical processes is underestimated in our model while iron fertilization effect is overestimated. Referee#2 questioned our stew more implicitly by asking question *"how do we know it [recipe] is the right one?"* and also suspected that in our model the iron fertilization is *"a sort of 'magic bullet' for drawing down carbon into the ocean"*. Indeed, the choice of the "carbon stew" is important for successful simulations of glacial cycles but only one among many other critical "modeling choices". The aim of our paper is not to present the ultimate solution for the "carbon stew" problem since at present this is simply impossible. The aim of our paper is to demonstrate that with a reasonable representation of physical, geochemical and biological processes in the model, it is possible to reproduce main features of Earth system dynamics over the past 400 kyr, including the magnitude and timing of climate, ice volume and $CO_2$ variations. The key world in the previous sentence is "reasonable". In a number of previous publications we have demonstrated that, in spite of its relative simplicity and coarse spatial resolution, CLIMBER-2 has a reasonable climate sensitivity ($3^{o}C$) and its spatial and temporal patterns of response to $CO_2$ and orbital forcing are in good agreement with the state-of-the-art climate models. Since both referees are concerned primarily about the "carbon stew", below we argue that our "carbon stew" is also reasonable and consistent with numerous studies published over the recent years.

The role of physical effects in glacial $CO_2$ drawdown. CLIMBER-2 is a rather simple and coarse-resolution model compared to the state-of-the-art Earth system models. This however, does not imply that it should necessarily underestimate (or overestimate) something. Unfortunately, Referee#1 did not explain why he/she believes that CLIMBER-2 underestimates contribution of physical processes to $CO_2$ drawdown and what is the correct value for this contribution. In Brovkin et al. (2007) we have shown that the net effect of the physical processes (solubility, ocean circulation, stratification, sea ice, but not changes in the global ocean volume and salinity) at LGM is about 45 ppm of $CO_2$ drawdown. This is not a small effect and we are not aware about results of 3-D ocean carbon cycle models which have much more. The last IPCC AR5 report summarized effect of different factors on glacial $CO_2$ and gave the median values of 25 ppm both for temperature and circulation effects. More recently, Buchanam et al. (2016) reported the total effect of temperature and circulation to be 40 ppm while Menviel et al. (2012) attributed only 20 ppm to physical processes. Kobayashi et al. (2015) found 45 ppm LGM drawdown, primarily through the physical processes. Thus we see no reason to assume that CLIMBER-2 underestimates effect of physical processes on glacial $CO_2$ drawdown. It has to be noted that the P-experiment, described in Brovkin et al. (2012), includes together with other physical processes also effect of the ocean volume change which counteracts other physical effects by ca 15 ppm. When 15 ppm are added at LGM to the results of P-experiment, the $CO_2$ drop at LGM becomes very close to 45 ppm reported in Brovkin et al. (2007).

Iron fertilization effect. We are surprised by the fact that both referees are so skeptical about importance of this mechanism. Since Martin's paper published 1990 (the paper was cited more 1000 times), the iron fertilization as one of plausible mechanisms of glacial $CO_2$ drawdown has been supported both by numerous modelling and paleoceanographic papers (e.g. Jaccard et al., 2016). As seen in Fig. 8c, at LGM the iron fertilization mechanism is responsible for ca. 25 ppm of $CO_2$ drawdown in our model. Note, that this number includes also effect of carbonate compensation. This value is well within the range of recent modelling estimates. For example, recent study by Lambert et al. (2015) attributed ca. 20 ppm to iron fertilization. Buchanan et al. (2016) attributed 55 ppm to the total change in biological pump. For comparison, if we sum up effects of iron fertilization and temperature-dependent remineralization depth, we arrive to less than 40 ppm. Schmittner and Somes (2016) used $^{13}C$ and $^{15}N$ isotopes to better constrain contribution of different factors to the LGM $CO_2$ drawdown. They came to the following conclusion: "Our results support Martin's [1990] hypothesis that increased iron input enhanced glacial ocean carbon storage by accelerating phytoplankton growth rates, consistent with previous studies [Bopp et al., 2003; Brovkin et al., 2007; Tabliabue et al., 2009]" and Galbraith and Jaccard (2016) arrived to a similar conclusion. Based on reasonable assumptions that are consistent with qualitative proxy evidence for the carbonate ion and oxygen concentrations, Anderson et al. (2015) concluded that about half of the DIC increase in the deep ocean during LGM had a respiratory origin.

LGM time slice versus transient simulation of glacial cycles. Most of previous studies concentrated on explaining 80 ppm $CO_2$ drop at LGM. Although the LGM "carbon stew" still remain a hot topic, it became also clear that very different combinations of numerous factors can explain 80 ppm drawdown. This is why recent development, first of EMICs and now of complex Earth system models, offers a new opportunity to better constrain "carbon stew" by performing transient simulations over the entire glacial cycle or, as in our case, even several glacial cycles. As we have shown in Brovkin et al (2012), at different phases of glacial cycle, the relative role of different factors differs significantly. This is why a good match between simulated and observed $CO_2$, not only during LGM but during entire 400 kyr of simulation, gives higher confidence that our 'modeling choices' are reasonable. Referee#2 suggested that our 'success' is almost solely explained by arbitrary tuning of iron fertilization effect which play the role of 'magic bullet' in our model. This is obviously not true. Fig. 8d clearly shows that iron fertilization explains less than 10 ppm during 80% of the last 400 kyr. At the same time, the agreement between observed and simulated $CO_2$ during these 80% is at least as good as during 20% when iron fertilization plays more significant role.

The use of paleoclimate data to constrain the carbon stew. Needless to say that paleoclimate proxies are essential component of evaluating of results of paleoclimate modeling. However, we do not share optimism of Referee#2 concerning possibility to constrain tightly the "carbon stew" by available paleoclimate data. Numerous attempts to achieve that (including the most recent by Schmittner and Somes (2016) and Heinze et al. (2016)), show a rather limited success. One of the reasons is that the proxy data syntheses are in the state far from being perfect, with proxies telling contradicting stories, such as Mg/Ca and organic proxies (eg. alkenons) for SST reconstructions. In spite of that we always tried to use available paleoclimate information to compare with modeling results. In the manuscript as well as in Brovkin et al (2007 and 2012) we showed and analysed a large

amount of oceanic and atmospheric characteristic such as atmospheric and oceanic $^{13}$C and $^{14}$C, oceanic oxygen, $CaCO_3$, etc. We will follow recommendations of the Referee#2 to make direct comparison with reconstructed oceanic $^{14}$C and atmospheric $^{13}$C.

Success or "success"? In his most general comment Referee#2 put the word *success* in quotes. We believe the quotes are unnecessary since our work indeed represents an important step forward in modeling and understudying of Quaternary climate dynamics. This is the first ever simulation of the past glacial cycle with the fully interactive ice sheet and carbon cycle models forced only by the orbital forcing. One should realize that dealing with long-term carbon cycle dynamics (volcanism, weathering, sedimentation) with geographically explicit Earth system is a very novel and challenging task, so one should not expect perfect agreement between modeling results and data. In the manuscript we thoroughly discussed all significant mismatches between data and model. Still the agreement between model and $CO_2$ and global ice volume is reasonably food.  For example, the correlation coefficients between simulated and modeling $CO_2$ is 0.86 in the one-way coupled experiment and 0.66 in the fully interactive. Root mean square errors (RMSE) are 13 ppm and 21 ppm respectively. For the last glacial cycle, for which model was calibrated, the agreement is even more impressive: correlation coefficients are 0.92 and 0.88 respectively; RMSE are only 11 and 13 ppm. Since the magnitude of stochastic millennial scale variability of $CO_2$ is about 10 ppm, such agreement is close to the upper theoretical limit.

Importance of our previous publications for understanding of GB17. Both referees complain that some important details of our modeling approach and analysis of mechanisms are not described in GB17. We will try our best to clarify as many issues as possible or to give proper references. However, it is important to realize that the manuscript presents results of the 20-years-long project and is based on a number of previous publications and it is both impossible and unnecessary to repeat things that we have published already. Fortunately, three of four  the most relevant papers needed for understanding of GB17, namely Brovkin et al. (2012), Ganopolski et al. (2010)  and Ganopolski and Calov (2010), are published in Climate of the Past and readily available for any potential reader. Only Brovkin et al. (2007) was published in Paleoceanography to which not everybody has free access.

**Response to Referee #1**

We thank the Referee #1 for useful and constructive comments. Please find our replies below.

*Ganopolski and Brovkin simulate four Glacial/interglacial (G/IG) cycles with the model of intermediate complexity CLIMBER2 in both a fully interactive and 1 way coupled mode. In both set ups, the model is able to reproduce the major features of G/IG cycles: i.e. changes in sea-level, ice-sheet extent (and volume), atmospheric CO2. . . It is an interesting study, certainly worth publishing in Climate of the Past. My main comment would be that I don't find the goals or conclusions of the study very clear.*

For a rather narrow community of fellow scientists striving to understand glacial cycles, the importance of successful simulation of glacial cycles with an Erth system model driven by orbital forcing alone is quite obvious. However, we agree that for a broader audience it is worth explaining why this problem is considered by some as the "holy grail" of paleoclimatology.

*The manuscript tries to tackle various aspects of G/IG cycles but without going deeply in any of them.*

We have commented on that in the general response.

*The authors are rightly very careful in not over-interpreting or making hasted conclusions from their results because the model used is quite simple.*

We fully agree that we used a rather simple model, although arguably the only one which is available at present for this sort of studies. However, it has to be noticed that complexity high resolution do not automatically resolve all problems because many processes in the Earth system are not yet properly understood.

*1. The first part of the introduction suggests that the radiative role of CO2 in driving 100kyrs G/IG cycles is controversial. To explore this, a simulation with constant pCO2 (240ppm) is performed. It leads to G/IG variations with 50% full G/IG amplitude and with dominant periodicity of 40ka. To me, this would tend to highlight the dominant role of CO2 in driving 100ka cycles, but this result or its implications are not really discussed.*

This is a misunderstanding. We do not downplay the role of $CO_2$ in amplifying of 100 kyr cycles. In Ganopolski and Calov (2011), the paper which is devoted to the nature of 100 kyr cyclicity, we wrote: "*the $CO_2$ concentration not only determines the dominant regime of glacial variability, but also strongly amplifies 100 kyr cycles*". What we stated in the introductio  is that we do not consider $CO_2$ as the **driver** of 100 kyr cycles, as some other workers proposed. According to our theory, 100 kyr cyclicity originates from the nonlinear response of the climate-cryosphere system to the orbital forcing through phase locking of long glacial cycles to 100 kyr eccentricity cycle (Ganopolski and Calov, 2011). In turn, 100 kyr cycles  are strongly amplified by $CO_2$. This result is consistent with the earlier findings of Andre Berger and colleagues. We will make this point more clear to prevent any possible misunderstanding.

*2. CO2 changes: The study simulates full G/IG changes in pCO2 due to a combination of processes and in global agreement with previous studies. However, due to the relative simplicity of the model and its configuration (zonally-averaged basin), I would think that the impact on pCO2 of oceanic circulation changes, sea-ice and wind related changes are underestimated, while iron fertilization changes are overestimated.*

This part of the comment we discussed in the general response.

*3. In addition, I am a bit surprised not to see any mention of the impact of changes in the carbonate system (e.g. shallow water carbonate deposition). A few studies (see A. Ridgwell or F. Joos studies) have shown that this has a significant impact on pCO2 particularly at the end of the deglaciations (early interglacial) and thus also glacial inceptions.*

The shallow water carbonate deposition is included in the CLIMBER-2 model which is described in Brovkin et al (2007) where we attributed to them 12 ppm of glacial $CO_2$ drawdown. We put more attention on this mechanism in our papers on interglacial simulations with CLIMBER such as Kleinen et al. (2016), Brovkin et al. (2016). We will mention the carbonate deposition mechanism in the revised manuscript.

*4. The authors highlight the impact of deglacial AMOC changes on the shape of the pCO2 trajectory at the end of the deglacial phase. This is an interesting aspect but:*

*i) Its reasons are not discussed in details*

It is true that we did not discuss this mechanism in GB17 but in Brovkin et al (2012) we devoted the entire section 3.3 to the discussion of millennial-scale variability in atmospheric $CO_2$ during the AMOC shutdowns. Now we introduced additional mechanism – temperature-dependent remineralization depth – which also contribute to $CO_2$ to the AMOC changes but to a smaller degree than the mechanism described in Brovkin et al. (2012). We will make this point clear in the revised manuscript.

*ii) Can we really believe it given that the shape and timing of the deglacial CO2 changes are not represented correctly and some processes are likely missing or misrepresented (e.g. shallow water carbonate deposition, oceanic circulation changes).*

We see no reasons why our results are not plausible. In fact, the shape and timing of glacial termination are not so bad, the shallow water carbonate deposition is accounted for and, as far as the ocean circulation changes are concerned, we know that at least for the LGM, CLIMBER-2 does a better job than many complex models (e.g. Weber et al., 2007; Muglia and Schmittner, 2015; Marzocchi and Jansenis, 2017). Indeed CLIMBER-2 correctly simulates shoaling of the glacial AMOC, decrease of deep Atlantic water ventilation and significant (above 1 psu due to continental ice sheets buildup) increase in salinity of the deep Southern Ocean water masses, while most of PMIP3 models show the opposite response. Marzocchi and Jansenis (2017) attributed this problem, at least partly, to the fact that most GCMs significantly underestimate sea ice extent in the Southern Ocean at LGM. At the same time, CLIMBER-2 (see Fig. 3 in Brovkin et al. 2007) simulates both modern and LGM sea ice extent in good agreement with modern and paleo data.

As far as 10-20 ppm $CO_2$ rise due to shutdowns of the AMOC are concerned, similar $CO_2$ rise simulated also in other models (e.g. Schmittner and Galbraith 2008; Matsumoto and Yokoyama 2013; Menviel et al. 2014, etc. ). And if some other models are unable to simulate such rise – this is their problem because 10-20 ppm $CO_2$ rise occurred in reality during most of Heinrich stadials and some non-Heinrich stadials.

Another important argument in favor of credibility of our finding is that it allows to understand why $CO_2$ overshoots coincide with strong overshoots in Antarctic temperature during MIS 5, 7 and 9, while during MIS 1 and 11 overshoots are absent both in $CO_2$ and Antarctic temperature records.

*Minor. Changes in weathering and its impact on pCO2 are not very clear. I realize it is mentioned in Brovkin et al., 2012, but maybe a brief description might be useful.*

We will add a brief discussion on simulated changes in weathering.

*Figure legends: Please make sure all appropriate references for the proxy are included in figure legends. For example Antarctic dust in Figure 2, Figure 4c. . . Proxy for atm. d13CO2 could be included in Figure 4b, even if they only cover part of the last G/IG cycle. Figure 8a: purple line.*

We will add references for proxies to the figure captions as suggested.

**Response to comments by Luke Skinner (Referee#2)**

We thank again Dr. Skinner for very detailed and useful review. Please find our replies below.

*My most general comment is that that study appears to focus overly on the 'success' of the numerical model simulations (and therefore the apparent success of the many model \*choices\* that have been implemented), rather than the justification or otherwise of the choices that have been made, for example as attested to by proxy data. In other words, it may well be that a viable 'recipe' for glacial-interglacial CO2 has been devised, but how do we know it is the right one?*

We responded to this comment and question in the General response.

*Arguably the only way to explore the latter question is to compare the biogeochemical/physical 'fingerprints' of that recipe with proxy data. My feeling is that more could (and probably should) be done in this regard, in particular with respect to carbonate chemistry, radiocarbon, oxygen and nutrient distributions/trajectories. Indeed, I would suggest that even if proxy data are too sparse to comprehensively test the particular 'CO2 recipe' that is adopted in this study (or if it is too much work to compile the data needed for this, since arguably this could be beyond the scope of this initial study), it should still be possible to identify its 'biogeochemical fingerprints' so that eventually the recipe we are being offered can be tested by others. Without this we are left without the means of assessing whether or not the CO2 recipe in this study is not only viable, but also possibly correct.*

*I would propose that three specific parameters to possibly consider in more detail are: radiocarbon, carbonate chemistry and oxygenation/respired carbon. Of these, radiocarbon and carbonate chemistry offer the best opportunities for data-model comparisons. I return to these suggestions below.*

This is, of course, a correct view on the model-data comparison, however, it might go beyond current state-of-the-art in both modelling and data. As we see it, few "robust" proxy-based features of the glacial states compared to interglacial ones are: (i) deep ocean (at least in Atlantic) was slower and colder; (ii) a biological productivity in the Southern Ocean was higher, however the deep ocean was not anoxic, and (iii) land had smaller or comparable amount of stored organic carbon. These 3 features are captured by our model. It is unclear for us whether regional details of proxy reconstructions are coherent enough to go beyond these three main features. To show regional details, we will provide additional plots on $^{14}C$, $^{13}C$, carbonate ion, and oxygen distributions in the ocean simulated by the model. For land, few data left beyond the last glacial maximum are coming from the pollen records, which is more qualitative than quantitative evidence for the land carbon storage.

Specific comments:

*1. The abstract states that the co-evolution of climate, ice-sheets, and carbon cycle have been simulated over 400,000 years using insolation as the only external forcing. This is an*

*impressive feat, and the reader wonders how this has been achieved of course; what are the key processes and feedback loops at the heart of the longstanding 'mystery of the ice ages'? It would be helpful if the abstract summarised the authors proposal succinctly.*

To learn more about the solution of 'mystery of the ice ages' the reader should read several our previous papers plus the paper on which one of the authors (AG) is currently working. Obviously, a comprehensive theory of glacial cycles cannot be presented in an abstract but we will do our best to accommodate this referee's suggestion.

*More specifically, it seems that a successful simulation of climate, ice volume and atmospheric CO2 has been achieved by appropriately scaling the rate of change of atmospheric CO2 to ice volume (using parameterizations for iron fertilisation and volcanic CO2 outgassing), and by further implementing additional climate-carbon cycle feedbacks that operate primarily through temperature-dependent respiration rates in the ocean, marine CO2 solubility effects and ocean circulation changes.*

Although this is rather a statement than a question or comment, we feel that we have to respond because this statement grossly underestimates amount of work we made over the past 20 years. Successful simulation of climate, ice sheets and $CO_2$ concentration is achieved not only (and mostly not) by scaling of something to ice volume but rather by the development, calibration and coupling of numerous models of individual components of the Earth system. Although carbon cycle is important for simulating of glacial cycles, climate and ice sheets are more important because glacial cycles can be simulated without carbon cycle model (with constant $CO_2$), while without climate and ice sheet components glacial cycles cannot be simulated.

As we already explained in the General response, half of glacial $CO_2$ drop in our model is explained by physical processes (solubility, stratification, sea ice, etc.) and this is why it is very important that CLIMBER-2 simulate changes in glacial circulation and deep water ventilation realistically (see response to Referee#2). All related climate-carbon cycle feedbacks are not "implemented" but are intrinsic part of our model, and they operate in our model the same way as in the most advanced ESMs. Some sort of scaling to ice volume is only applied to volcanic outgassing and iron fertilization and these two effects never give together more than 35 ppm.

*The extent to which the phenomena have been implemented as modelling choices, and the extent to which the magnitude of their impacts (e.g. on CO2) depends on parameter choices, should be made clear.*

Parametrizations for iron fertilization and volcanic outgassing do affect the magnitude of the atmospheric $CO_2$ changes, but even without them – and with constant $CO_2$ - the system will go through the glacial cycles, albeit with smaller amplitude. Of course, any modeling result depends on the choice of modeling parameters. We will make this point more clear in the revised manuscript.

*2. The abstract focuses on the deglaciation as being particularly sensitive to parameter choices, apparently in contrast to the rest of the glacial cycle (for which many features are argued to be 'rather robust'). I feel that this might be a little misleading; can the meaning of*

*this statement be clarified? In what sense exactly can modelled features be said to be robust?*

"Robust" here means that qualitative evolution of the system - such as direction of changes and occurrence of events - is not dependent on the choice of parameters, of course, within their plausible range. The $CO_2$ response to the AMOC shutdown is also robust in the model, however, the longer the shutdown, the stronger is an overshoot and the $CO_2$ recovery afterwards. In the $CO_2$ record, it looks like an overshoot and stabilization, like in the Eemian, or as small jump continued by increasing $CO_2$, as in the Holocene (Fig.2 , TI). As the timing of AMOC changes is very sensitive to the freshwater flux, these two types of responses could occur by chance, and therefore are not "robust".

*3. The issue of CO2 overshoot: this is highlighted in the abstract as a key finding, but it needs to be explained more fully I think. Why exactly does this phenomenon occur? Does it depend on model choices and if so which ones, or is it a fundamental aspect of the physics in the model?*

We described the mechanism of $CO_2$ response to AMOC shutdown in Brovkin et al (2012), section 3.3 and it is indeed related to fundamental aspect of physics and carbon cycle in the model.  Incorporation of the temperature-dependent remineralization depth additionally Contribute to $CO_2$ response to AMOC changes but the mechanism described in Brovkin et al (2012) remains the dominant one. We will make this point clear in the revised manuscript.

*It would appear that the AMOC is sensitive to freshwater forcing throughout the deglaciation, but that AMOC anomalies early in the deglaciation (and during the glacial?) have no appreciable carbon cycle impact; why is this?*

This is absolutely correct observation. Indeed, during periods of strong dust flux, response of $CO_2$ to the same AMOC changes is smaller compare to the experiment without iron fertilization. This is consistent with the fact that during Heinrich event 1 no significant changes in $CO_2$ occurred while during previous Heinrich events $CO_2$ rose by 10-20 ppm. Why this happens n CLIMBER-2 requires further investigation.  In any case, the influence of enhanced biological pump on $CO_2$ sensitivity to AMOC is not relevant for $CO_2$ overshoot at the end of glacial termination because iron fertilization ceased to influence $CO_2$ well before the end of glacial terminations.

*Is marine soft tissue pump efficiency 'maxed out' (exhausted) and therefore insensitive to further enhancement until the parameterizations for increased Fe-fertilisation and nutrient respiration rate are released?*

This argument is not clear since the AMOC shutdown and iron fertilization cause opposite effect on $CO_2$.

*More explanation is needed for this phenomenon, especially if it is highlighted as being particularly noteworthy.*

We agree and will explain in more details.

*4. Page 4, Line 11: the way in which iron fertilisation is implemented needs to be clarified. How is exactly is nutrient utilisation scaled with dust and on what basis? How do we know that the right scaling has been applied, or is it essentially arbitrary? Can the scaling be justified on the basis of nitrogen isotopes (simulated) or anything else? Without such details the iron fertilisation mechanism will always seem like a sort of 'magic bullet' for drawing down carbon into the ocean.*

The nutrient utilization is linearly proportional to the amount of "Antarctic dust" which is prescribed from Antarctic ice cores in the case of one-way and computed from sea level in the case of fully coupled experiment. Nutrient utilization has upper limit corresponding to the complete utilization of phosphates at the surface. The iron fertilization plays a role only during the 2nd part of the glacial cycles when global ice volume is large than 50m. We will describe this parameterization in the Appendix.

*5. Page 4, Line 23: the implementation of radiocarbon in the model should be explained a little more clearly too (e.g. is it simulated as an isotope tracer that undergoes gas exchange, fractionation etc... or is it a pseudo-tracer with a decay timescale that is restored to a particular value at the ocean surface?). Note that Hain et al. (2014) did not produce a radiocarbon production scenario; please check this reference (ultimately the production scenario will be based on Be-10 or geomagnetic field strength and the original references should be cited). In general I think that more should be made of the radiocarbon outputs, e.g. in comparison with existing data. Such data should be added to figure 12 for example, and any agreement/disagreement discussed. I return to this later.*

Indeed, we apologize for not citing original $^{14}$C production model used by Hain et al: Kovaltsov, G.A., Mishev, A., Usoskin, I.G., 2012. A new model of cosmogenic production of radiocarbon 14C in the atmosphere. Earth and Planetary Science Letters 337, 114-120. We used these data as provided by Hain et al. and scale them for pre-industrial state assuming that the system is in equilibrium, i.e. that production is equal to decay in the model. As a result of the scaling, the atmospheric $\delta^{13}$C at 0 ka is around 0 permil, as in the IntCal data.

*6. Page 5, Line 5: notably this way of doing greenhouse gases will produce incorrect results for millennial timescales, since methane and CO2 are not in phase during DO/ Heinrich events. Does this matter; can it be shown that it does not matter?*

The magnitude of $CH_4$ changes during DO events is typically less than 150 ppb that represents only 5% of the radiative forcing of all GHGs during glacial cycles. Since periodicity of DO event is much shorter than the orbital time scales, DO event represent nothing more than a red noise of a small magnitude and it cannot produce any measurable effect on glacial cycles. On the other hand, even if one would have a model which incorporates methane cycle and is able to simulate DO events rather realistically, the right timing of DO events cannot be simulated anyhow because they are random. Thus 5% errors in instantaneous GHG forcing on millennial time scale is both unavoidable and insignificant.

*7. Page 5, Line 22: can this careful calibration of volcanic outgassing be tested against e.g. atmospheric d13CO2 for example (note that these data are available for the last glacial cycle from Eggleston et al. 2016)? If the volcanic control on atmospheric CO2 is so strong, it might*

*also be expected to affect the isotopic composition of the atmosphere quite strongly (as well as the deep ocean carbonate system - more on this later). Is the surface (i.e. non-solid Earth) carbon cycle balanced; i.e. is 5.3TmolC/yr going back into the solid earth in the model? All of these are important questions that jump out at the reader, but are not dealt with at all in the current manuscript.*

The volcanic $^{13}$C is assumed to be constant (2 permil); its isotopic footprint is similar to the carbonate footprint that goes out of the system. Therefore, the effect of volcanic outgassing on the atmospheric $^{13}$C is negligible on the timescale of simulations. The carbon budget was balanced for the preindustrial simulations as in Brovkin et al. (2007, 2012) – the silicate weathering of 12 Tmol was balanced by ½ of it with 6 Tmol of volcanic outgassing. For glacial cycles, silicate weathering is changing depending on the runoff, so the average volcanic outgassing used in glacial simulation should be slightly less than in the pre-industrial state.

*8. Page 5, Line 30: this procedure for 'initial condition conditioning' is very interesting, but it is not so obvious why the system should converge on the same initial and final states, regardless of the history of evolving boundary conditions over 410kyrs; is it possible to clarify? What component is drifting that depends on the state of the system (and that eventually reaches an equilibrium through this iterative process)?*

The long-term carbon cycle (outgassing, weathering, sedimentation) requires a fine balance. With smaller or higher carbon input, the system will drift either up or down, but after some time will find a new cycling state with higher or smaller $CO_2$ level.  Therefore the model parameters should be properly tuned and initial conditions are selected in a way to prevent such drift. The main quantity which has to equilibrate is the total ocean carbon content.

*9. Page 6, Line 30: the lag of CO2 is an important clue as to what is (perhaps) not right in the model parameterizations that have been selected. One wonders if this has something to do with the choice to scale iron fertilisation with ice volume: dust does not track sea level very closely in reality, and more specifically it drops off rapidly before sea level has risen much in the deglaciation. Or is the lag due to something else? More analysis of the source of this mismatch would be illuminating (more illuminating than if the model happened, perhaps accidentally, to match observations perfectly).*

We fully agree that any mismatch between model and data indicate that something is not perfectly right in the model. In this specific case, this is definitely not related to the parameterization of iron fertilization. Since both in the one-way coupled experiment and in the fully interactive experiments the dust flux drops nearly to zero already during the initial phase of glacial terminations, the rapid decline of the biological pump facilitate rapid  $CO_2$ rise during the initial phase of deglaciaton. In fact one is potential candidate for creating this problem is the land carbon which starts to grow rapidly in parallel with ice sheets retreat. This is confirmed by the experiment in which land carbon is not accounted for and in which the lag between simulated and observed $CO_2$ is somewhat smaller. However because the lag is still present even in this experiment, it must be other problems. For obvious reason we cannot say what is wrong because if we would know, we would fix the problem and obtain perfect results.

*10. Page 6, Line 32: as noted above, a more detailed explanation is necessary for the*

*mechanisms underlying the overshoot in CO2, and for CO2 release as a function of AMOC variability in general. There is not universal agreement amongst models for millennial scale controls on atmospheric CO2 and the role of the AMOC, so it will be useful to know what is going on in this particular model experiment, and why the carbon cycle response to AMOC changes is so context dependent. On page 9 it is suggested that the CO2 overshoots depend primarily on remineralisation depth changes that in turn stem from subsurface heat anomalies, but this is not clearly stated or explored anywhere else.*

See our response to the comment N3.

*11. Page 7, Line 20: again, this lag, and it's increase in the fully coupled runs is important, and should be diagnosed more clearly, as it is telling us something important about the model choices that have been implemented.*

Increase of the lag between simulated and observed $CO_2$ in the fully coupled experiment does not provide additional information about the problems with the carbon cycle model. In the fully coupled simulations, where strong positive feedbacks between $CO_2$, climate and ice sheet are activated, any errors already present in the one-way coupled experiment will be strongly amplified. This is why accurate simulation of climate, ice sheet and $CO_2$ in the fully interactive simulations is much more challenging task comparing to simulations with prescribed $CO_2$.

*12. Page 8, Line 4: it is very interesting and important that CO2 changes on a dominantly 100ka timescale are not needed to produce glacial cycles in the model, but where does the 100kyr timescale for ice sheet growth/decay come from in this model;*

This is described in Ganopolski and Calov (2011) paper which is entirely devoted to the nature of 100 kyr cyclicity. In this paper we demonstrated that the 100 kyr cyclicity originates from the nonlinear response of the climate-cryosphere system to the orbital forcing through the phase locking of long glacial cycles to the 100 kyr eccentricity cycle.

*is it simply the timescale at which the ice sheets get big enough for the dirty-ice albedo instability to kick in?*

It is not simple. The time scale of ice sheets is about 30 kyr (Calov and Ganopolski, 2005), which is much shorter than 100 kyr but much longer than half of precessional cycle and it takes several precessional cycles for ice sheets to reach their "critical" size after which termination becomes possible. Most favorable conditions for reaching of this critical size is the periods of low eccentricity when one of positive precessional cycle coincide with a negative obliquity cycle. This is why long glacial cycles are phase locked to 100 kyr eccentricity cycle (Ganopolski and Calov, 2011). In turn, critical size is related not only to positive dust feedback but also to several other processes and feedbacks. This issue will be further discussed in a forthcoming paper.

*If so, how is that feedback constrained (is the time scale a model choice once again or is it due to a fundamental limitation on ice growth rates and basal sliding etc...);*

The time scale of ice sheet response to the orbital forcing is not prescribed and therefore it

is not "a model choice". The time scale of ice sheets response to orbital and other climate forcings is determined by surface mass balance and ice sheet dynamics. We simulate surface mass balance using a physically based energy balance approach which has been successfully validated against present day observations and other models. Our ice sheet model SICOPOLIS is the standard 3-D thermomechanical model. This model also has been extensively tested for present day and paleo ice sheets. The basal sliding is parameterized in SICOPOLIS the same way as in other similar models. As any parametrization, it is a simplification and there are uncertainties but a good agreement between simulated and reconstructed ice sheets during the last glacial cycles gives as confidence in our model.

*how do we know it should happen on that timescale?*

It is not clear what is meant under "it". If this means 100-kyr time scale, then as explained above, this time scale is not directly related to the time scale of ice sheets.

*13. Page 8, Line 24: can it be stated that the 'better' performance of the enhanced freshwater flux experiments indicates an under-representation (or misrepresentation) of the role of ocean circulation perturbations, at glacial transitions in particular?*

This is somewhat strange interpretation of the fact that the experiment with 10% enhanced freshwater flux has a "better" performance. First, it is only marginally "better". The RMSE of $CO_2$ in ONE_1.1 is 13.4 ppm, while in ONE_1.0 RSME is 14.9 ppm (see also Fig. 2e). Second, in reality the Northern Hemisphere ice sheet volume at LGM is not even known with accuracy of 10%. Therefore both experiments can be considered as equally plausible.

*Is it possible that it could also be that this enhanced forcing is needed to compensate for other biases, e.g. from iron fertilization or volcanic CO2 parameterizations? How would we know, what do we learn from this?*

First, we do not agree that model biases originate from iron fertilization or volcanic outgassing parameterizations. To the contrary, these two processes are introduced into the model to reduce model biases. Second, we cannot see any relationship between sensitivity of the AMOC to freshwater flux and iron fertilization because differences between these two experiments are only seen at the end of glacial terminations (Fig. 2e) when iron fertilization does not play any role. The only thing which one can learn from comparison of these two experiments is that the timing of AMOC resumptions at the end of glacial termination is very sensitive to the magnitude of freshwater flux. However, this is not surprising – it has been shown already in Ganopolski and Roche (2009).

*14. Page 9, Line 1-7: it would be helpful to include a table that clarifies the 'carbon stew' and the contribution of each mechanism that is implemented, e.g. based on average glacial and interglacial values.*

We will include a table with contribution of each mechanism for several time slices based on factorial experiments.

*15. Page 9, Line 14: the original references of Matsumoto (2007) and Matsumoto et al. (2007) are missing here, and where the notion of temperature dependent respiration rates is*

*introduced.*

Thank you, the original references will be added.

*16. Page 9, Line 25: some more detail on the volcanic CO2 implementation is needed; what about the balance of marine versus sub-aerial volcanism, and their different responses to ice vs water loading*

We do not distinguish between these two sources because at the orbital time scales their effect on atmospheric $CO_2$ is nearly identical

*… how is this treated and on what basis is a particular magnitude of volcanic CO2 flux chosen?*

The values of the parameters in the equation for volcanic gas outgassing (p. 17) is chosen to produce glacial-interglacial variations in volcanic outgassing of about 30% of its average value which adds additional 10 ppm to glacial $CO_2$ drawdown.

More justification/testing of the volcanic CO2 implementation is also needed;

Again, we should repeat that the aim of the paper is not to justify or test individual components of the carbon stew - this has been done already in numerous papers, including our own. As far as volcanic outgassing is concerned, there is a number of papers which argues in favor of this mechanism such as Huybers and Langmuir (2009), Lund et al. (2016), Huybers and Langmuir (2017) and many others. This mechanism has been tested already with a carbon cycle model by Roth and Joos (2012) who concluded that a large change in volcanic outgassing during termination I cannot be ruled out but it occurs too late to be the main cause of deglacial $CO_2$ rise . Of course, we do not consider volcanism to be the main cause of deglacial $CO_2$ rise. Note that Roth and Joos considered much more drastic scenarios, where volcanic outgassing increased by factor 2 and more (in GB17 volcanic outgassing change by only 30%) during glacial termination. As the results, additional $CO_2$ rise due to including of time-dependent volcanism in Roth and Joos (2012) ranges between 13 and 142 ppm while it is less than 10 ppm in our case.

*what is the impact on marine carbonate chemistry and does this tally with proxy evidence (it should cause marine carbonate ion concentrations to go up in the glacial, at odds with data from the Atlantic where it goes down, and the Pacific where it stays pretty constant)?*

The impact is essentially nil. The magnitude of present volcanic outgassing is about 0.1 GtC. Our parameterization introduces anomaly of about ±0.015 GtC while the total ocean carbon content is about 40,000 Gt. By comparing these two numbers, it is obvious that at the orbital time scales $10^4$-$10^5$ yrs the impact of variable volcanic outgassing cannot affect marine carbonate chemistry.

*Is there a longer-term feedback via carbonate preservation*

This feedback always operates in our model irrespective of the source of carbon

*are changes in volcanism perfectly balanced by weathering and sedimentary carbon outputs in the model, and if not what is compensating for the drift in global 'surface' carbon inventories that would result from this?*

Averaged over long period of time (> 100 kyr) volcanic outgassing must be balanced by weathering and sedimentation, otherwise the model will drift away from the realistic state. This is why we tune the value of average volcanic outgassing to prevent such drift.

*Also, as noted above, please state what the impacts of the changing volcanic carbon fluxes on atmospheric carbon isotopes are: are they essentially nil?*

Yes, as has been shown already by Roth and Joos (2012), it is essentially nil.

*17. Page 10, Line 11: it is stated that the brine rejection parameterization cannot be tested with observational data, but is this entirely true/fair, especially given the lack of testing offered in this study for the volcanism and temperature dependent respiration rate mechanisms?*

Indeed, both variable volcanic outgassing and brine rejections are hypothetical mechanism with some support from paleodata. Temperature-dependence of organic matter decomposition is well established process, so we have a higher confidence in this process than in brine rejection or volcanism. However, as it is explained on page 10 in GB17, the fact that it is unknown whether efficiency of brine rejection can be close to 100% during glacial time as postulated in Boutess et al. (2012). Even more serious problem is that the temporal evolution of this key parameter is unknown. Boutess et al. (2012) and Mariotti et al. (2017) assumed rapid drop in this value from maximum to zero at the beginning of glacial termination which is hard to justify in a view that the Antarctic ice sheet did not start to retreat at that time. Interestingly, Menviel et al. (2012) who also tested the role of brine rejection, assumed a totally different temporal scenario for the brine rejection, with the maximum of brine rejection efficiency reached in the middle of glacial cycle and essentially zero at LGM (their Fig.2 and Table 2). It is important to stress that the main strength of our modeling approach is that we do not use any explicitly time-dependent model parameters. Only orbital forcing is prescribed in the fully interactive run and the rest our model does on its own. Therefore we cannot use the approach by Boutess et al. (2012) and Mariotti et al. (2017). Until a clear idea of how to relate brine rejection efficiency with the simulated state of the Earth system will emerge, we simply cannot introduce brines in our "carbon stew".

*A critical analysis of **all key modelling choices** should be provided; not just for brine rejection.*

Unfortunately, this is impossible. Earth system models are based on numerous modeing choices which are crucial for successful simulation of glacial cycles. After all although CLIMBER-2 is an EMIC, it is still incomparably much more complex model than for example box-models and its program codes consist of more than 30,000 FORTRAN lines.

As far as the composition of "carbon stew" is concerned, individual processes have been already analysed in our previous papers and numerous papers of other authors. Even a rather exotic, volcanic mechanism, has been tested already by Roth and Joos (2012). As far as the iron fertilization and temperature-dependent remineralization depth are concerned,

they are now routinely implemented in ocean carbon cycle models.

There are two reasons why we specifically addressed the brine rejection mechanism in GB17. First, we implemented this parameterization in CLIMBER a while ago but never described and tested it. Second, we were particularly interested in whether this parameterization helps to resolve the problem with atmospheric [14]C and we believe that the results presented in the section 5.2 of GB17 are worth discussing.

*18. Page 11, on deglacial d13Catm: The text gives the impression that the deglacial d13Catm tends are quite accurately reproduced, but the match is not great. The 'W' in deglacial d13Catm is not particularly clear; what is this mismatch attributable to? Does it mean that the model is not simulating the correct marine carbon cycle response to AMOC change? Also, why is the more substantial early Holocene d13Catm rise seen in available data not reproduced; does this mean that terrestrial carbon uptake is too small in the model? Do marine carbonate ion values confirm this latter possibility or not (or at least demonstrate that marine carbonate ion reconstructions could be used to test the model)? I think a great deal more should be made of the isotope simulations and their comparison with proxy data.*

We disagree that "*The text gives the impression that the deglacial d13Catm tends are quite accurately reproduced".* On page 11 we wrote that "the magnitude of the $\delta^{13}$C drop is in a good agreement with empirical data" which is correct. Then we wrote "The model is also able to simulate W-shaped $\delta^{13}$C evolution" which is also true. The problem is that modeled W-shape is shifted compare to the real one because model analog of Bolling-Allerod occurs ca. 1500 yrs earlier than in reality. This is absolutely natural because this event occurs in the model internally without any prescribed external forcing and therefore one cannot expect that it should occur at the same time as in reality. If we would shift the red curve in Fig. 9c, such a way that the timing of simulated warm event would be in agreement with the real Bolling—Allerod (which we will do in the revised manuscript), then the visual agreement is much better. Then we wrote on the same page that "At the same time, simulated present-day atmospheric $\delta^{13}$C is underestimated compare to ice-core data by ca 0.2‰" which is not perfectly correct because in reality this difference is even smaller. The reason for this data-model mismatch during Holocene is not clear. The total land carbon uptake in the model during deglaciation is larger than 3000 GtC (see Fig. 13a), which is in an agreement with current estimates. However, in our model, atmospheric $\delta^{13}$C is much stronger controlled by the marine processes than terrestrial one. We therefore assume that $\delta^{13}$C mismatch reflects mismatches in the ocean C cycle. There are few reconstructions of the carbonate ions available; as in Brovkin et al. (2012), the model simulations of $CO_3^-$ are qualitatively in line with observations. We rely more on $CaCO_3$ sedimentation records which show enhanced preservation during deglaciations, when deep ocean carbon was released to the atmosphere and deep waters became less acidic. This spike in preservation is reproduced by the model.

*19. Page 12, on deglacial 14C: Even more so than for the stable carbon isotopes, I think that a great deal more should be done with the radiocarbon simulations and their comparison with observations. Figure 10 should really include data, as should Figure 12 (this could be made substantially easier to include by a recent compilation by Skinner et al., Nature Communications, 2017).*

We are grateful to the Referee for pointing on his recent paper. We agree that [14]C data are

useful for testing the model and we are going to use $\delta^{14}R$ vertical profiles from fig 3 in Skinner et al. (2017) for comparison with our profiles in Fig. 12. As far as the Referee's suggestion to plot individual data points in our fig. 10c is concerned, we have doubts. The data are too noisy. For example in the deep Atlantic between 20N and 40N radiocarbon ages are scattered between 1000 and 3000 years (Fig. 4a in Skiner et al., 2017) which suggests that either the uncertainties of radiocarbon age as high as 50% or that the data contain strong regional signal which anyhow cannot be reproduced by the zonally averaged ocean model. Instead we will make a qualitative comparison between our Fig. 10 and fig. 4 from Skinner et al. (2017).

*Radiocarbon data provide very strong constraints on the ocean state; if the simulation does not fit the available data, some discussion is warranted. This relates to the following section, where it emerges that the model simulation not preferred by authors, using brine rejection as a stratification mechanism, produces radiocarbon data that better fit the data (though again, no direct comparison with data is shown).*

By comparing our fig. 12 with fig. 3 from Skinner et al. (2017) we cannot understand why the Referees arrived to such conclusion. At LGM our standard model (without brines, blue) gives 2000-2500 yrs in the deep Atlantic and 2500-3000 yr in the deep Pacific which is in very good agreement with Skinner et al. (2017). At the same time, the model with brines (green) gives age more than 3000 yrs in the deep Atlantic and 3500-4000 yr in deep Pacific which is much older than in Skinner et al. (2017).

*20. Page 12 , Line 14: it is stated that the radiocarbon data are in good agreement with Roberts et al. (2016); however that publication did not present radiocarbon data. Please correct the reference and/or clarify.*

The reference will be corrected

*21. Page 12, Line 28: if the preferred model simulation does not fit the radiocarbon observations, does this not mean that the "CO2 stew" proposed in the manuscript must not be completely accurate? Please clarify.*

Of course, there is no guarantee at all that the magnitude and timing of mechanisms proposed in the paper are quantitatively accurate. On the other hand it is unclear whether there is a direct link between the composition of the "carbon stew" and LGM radiocarbon problem. We can only discuss qualitative fit of the model to the $^{14}C$ and other proxy data, and suggest possible reasons for disagreement.

*22. Page 12, Line 30: in the manuscript DD14C is used as the preferred ventilation metric; however, this metric does not scale with the isotopic disequilibrium between two reservoirs in a constant manner. In other words, a given DD14C value will reflect a different degree of isotopic disequilibrium (or ventilation age) depending on the absolute D14C. This not only makes DD14C a particularly confusing metric, but it also means that simulated DD14C values can match observed values without being correct if the absolute atmospheric/marine D14C values are too high/low. This indeed seems to be the case here, as the simulated D14Catm at the LGM is _150 permil lower than observed. For these reasons I would urge the authors to use marine vs atmosphere radiocarbon age offsets (B-Atm), which can also be converted to a*

*ratio of isotopic ratios (or F14b-atm, Soulet et al., 2016) if a semblance of 'geochemicalness' is required.*

We are grateful to the Referee for this suggestion and will show the radiocarbon age instead of $\Delta\Delta^{14}$C in all figures.

*23. Page 13, Section 5.3: can the authors state clearly what the implications are, if there are any, for marine and atmospheric carbon isotopes (13C, 14C) of the terrestrial carbon shifts, e.g. at the last deglaciation? It has been proposed that parts of the observed deglacial 14Catm record might be explained by permafrost changes; do the model results support a significant impact on deglacial atmospheric radiocarbon (or d13C)?*

We investigated the permafrost carbon hypothesis and found that its impact on $CO_2$ is not significant enough to explain trends in $^{14}$C and $^{13}$C. The land surface model operates with large grid cells which smooth out possible abrupt changes in the land C storage.

*24. Page 13, Line 33: "..the model simulates the correct timing of glacial terminations..." I would suggest to be more precise (e.g. ice volume, but not CO2?), and perhaps to quantify this as being within a certain (millennial?) margin of error.*

Why "not CO2"? Glacial terminations occur roughly every 100 kyr. The lag of simulated $CO_2$ relative to the observed one (measured by timing of termination midpoints) is ca. 3 kyr. Even if the ice core $CO_2$ age is perfectly correct, our model computes terminations with the accuracy 3% which is a very high accuracy by the standard of climate modeling. In the revised manuscript we will provide quantitative information on how accurately our model simulates glacial inceptions.

*25. Page 14, Line 2: "...ocean carbon isotopes evolution is in agreement with empirical data." Should stable carbon isotopes be specified; should the statement be qualified somewhat (e.g. global spatial patterns have not been matched.. and the fit is assessed only in very general terms)?*

Thank you, we'll specify that the match is for stable carbon isotopes.

*26. Page 14, Line 3: should this read "the magnitude of atmospheric 14C change is underestimated"?*

Yes, this is correct

*And on Line 5, I would say that the statement regarding disagreement with data has not really been backed up very strongly as there is no illustration of a comparison with data in the manuscript.*

We will present such comparison in the revised version of the manuscript which fully support our statement.

*27. Page 14, Line 10: I think that some more explanation is required for what is meant by 'robust' in this context.*

See reply on our understanding of robustness above. We will add a comment on it into revised manuscript.

*28. Page 16, Line 25: as noted above, the scaling of iron flux with sea-level is arguably questionable, since although dust fluxes in Antarctica increase relatively late, when sea level has fallen and CO2 has already dropped somewhat, it is also true that dust fluxes drop off very quickly on the deglaciation, before sea level as risen appreciably. Does this not mean that the 50m RSL threshold for dust changes is somewhat incorrect (i.e. it has the effect of keeping iron fertilisation strong for too late in the deglaciation)?*

First, parameterization of the dust flux is only applied in the fully interactive experiments. In the one-way coupled experiment, the dust flux is taken from the ice core data and it drops rapidly at the beginning of each termination. Second, in the parameterization described on page 16, the dust is not just scaled with sea level, it has a much more complex dependence on sea level, and time derivative of sea level dS/dt plays crucial role. As the result, after the LGM the term dS/dt turns negative and dust flux starts to decline almost immediately after the LGM and not after crossing of 50m threshold. This formula was chosen by tuning simulated dust to the measured in ice cores. As one can see from Fig. 5c, the agreement between simulated and measured dust is not bad. In any case, there is no tendency for simulated dust to stay too high too long during glacial terminations. In fact, the effect of iron fertilization on $CO_2$ in the second half of glacial terminations is always small. Therefore Referee's concern that our parameterization keeps "*iron fertilization strong for too late*" is not justified.

*A plot of how the timing of dust/iron fluxes in the model compare with the timing of dust fluxes in Antarctic ice cores might provide a test of this. I would suggestion including such a figure as a justification of the chosen parameterization.*

This is done already. See Fig. 5c in GB17.

*Again, I think that a clear description is needed for how export production is scaled to dust fluxes in the model, and on what basis the chosen scaling is justified (it would be nice to know what the Southern Ocean and global export productivity is in the model on average for glacial and interglacial states). How is iron release from dust simulated, how is biological activity as a function of iron availability simulated etc..? I think that a clear description of how biological carbon fixation/export is linked to dust fluxes should be included in the appendix.*

The effect of iron fertilization in our model is highly parameterized. This parameterization will be described in the Appendix.

*29. Figure 4: atmospheric d13C data for the last glacial cycle and deglaciation should be added, including e.g. Eggleston et al. (Palaeoceanography, 2016).*

Thank you, we will add comparing simulated $\delta^{13}$C with the Eggleston et al. (2016) reconstruction, as well as a brief discussion of possible reasons for mismatches.

*30. Figure 7b: perhaps add the power spectrum for a appropriate insolation record, as a dashed line?*

We will add the spectrum of insolation to the figure

*31. Figure 8: I personally would find it useful if the plots b-e were drawn as filled curves, either side of the zero line, so that it was clear when each process was acting as a source or sink for CO2.*

This is a good suggestion. We will add the zero lines to the figure 8.

*32. Figure 9: I think this figure would benefit from adding a comparison between simulated and observed marine radiocarbon ventilation ages some key locations/regions. It may provide insights into why the atmospheric simulations do not match the observations.*

Marine radiocarbon ventilation age from individual locations cannot explain changes in atmospheric $^{14}$C. This only can be done by proper global averaging but paleodata are too sparse and uncertain to produce such global averaging.

*33. Figure 10: why do the plots only go to 40°S? I think this figure would greatly benefit from added data comparison. For this it would be essential to convert the radiocarbon activities to radiocarbon age offsets or radiocarbon ratios (i.e. not relative deviation offsets).*

See our response to the comment N19.

*34. Figure 11: probably it would be good to add an indication of what the green line is (even though it is obvious by process of elimination).*

 We will add the meaning of the green line to the figure caption.

*Does the brine rejection experiment not include freshwater pulses during deglaciation; why does it not exhibit any deglacial anomalies at all? Again, data might usefully be added to the figure for comparison.*

The experiment with brine rejection has almost the same freshwater forcing as the standard run. However, intensive brine rejection in combination with density-dependent vertical mixing strongly affects density fields, ocean circulation and its sensitivity to freshwater flux. As the result, the model with brines does not simulate millennial scale variability during termination.

*35. Figure 12: this figure is the most obvious one in which to include a comparison with observations, along with an addition to the text of a discussion of any mismatches between the various experiment outputs and the observations. It seems to me that if the simulation does not fit the data, then something is amiss, which we might learn from if it was identified.*

We will add observational data from Skinner et al (2017) to this figure and will discuss "mismatches".

*36. Figure A1: What are the different coloured substrates? Perhaps more can be done with this figure?*

We will expand the figure caption making it self-explanatory.

We will also account for all minor points in the revised manuscript.

**References**

Anderson, R.F., Allen, K.A., Yu, J., and Sachs, J.P. (2015) Ocean Stratification, Carbon Storage, and Calcite Compensation throughout the Late Pleistocene Glacial Cycles, Nova Acta Leopoldina NF 121, Nr. 408, 23 –27.

Bouttes, N., Paillard, D., Roche, D. M., Waelbroeck, C., Kageyama, M., Lourantou, A., Michel, E., and Bopp, L.: Impact of oceanic processes on the carbon cycle during the last termination, Clim. Past., 8, 149-170, 10.5194/cp-8-149-2012, 2012.

Brovkin, V., Ganopolski, A., Archer, D., and Rahmstorf, S.: Lowering of glacial atmospheric $CO_2$ in response to changes in oceanic circulation and marine biogeochemistry, Paleoceanography, 22, 14, 10.1029/2006pa001380, 2007.

Brovkin, V., Ganopolski, A., Archer, D., and Munhoven, G.: Glacial $CO_2$ cycle as a succession of key physical and biogeochemical processes, Clim. Past., 8, 251-264, 10.5194/cp-8-251-2012, 2012.

Brovkin, V., Bruecher, T., Kleinen, T., Zaehle, S., Joos, F., Roth, R., Spahni, R., Schmitt, J., Fischer, H., Leuenberger, M., Stone, E. J., Ridgwell, A., Chappellaz, J., Kehrwald, N., Barbante, C., Blunier, T., and Jensen, D. D.: Comparative carbon cycle dynamics of the present and last interglacial, Quat. Sci. Rev., 137, 15-32, 10.1016/j.quascirev.2016.01.028, 2016.

Buchanan, P. J., Matear, R. J., Lenton, A., Phipps, S. J., Chase, Z., and Etheridge, D. M.: The simulated climate of the Last Glacial Maximum and insights into the global marine carbon cycle, Clim. Past., 12, 2271-2295, 10.5194/cp-12-2271-2016, 2016.

Calov, R., and Ganopolski, A.: Multistability and hysteresis in the climate-cryosphere system under orbital forcing, Geophys. Res. Lett., 32, 4, 10.1029/2005gl024518, 2005.

Calov, R., Ganopolski, A., Claussen, M., Petoukhov, V., and Greve, R.: Transient simulation of the last glacial inception. Part I: glacial inception as a bifurcation in the climate system, Clim. Dyn., 24, 545-561, 10.1007/s00382-005-0007-6, 2005.

Eggleston, S., Schmitt, J., Bereiter, B., Schneider, R., and Fischer, H.: Evolution of the stable carbon isotope composition of atmospheric $CO_2$ over the last glacial cycle, Paleoceanography, 31, 434-452, 10.1002/2015pa002874, 2016.

Ganopolski, A., Calov, R., and Claussen, M.: Simulation of the last glacial cycle with a coupled climate ice-sheet model of intermediate complexity, Clim. Past., 6, 229-244, 2010.

Ganopolski, A., and Calov, R.: The role of orbital forcing, carbon dioxide and regolith in 100 kyr glacial cycles, Clim. Past., 7, 1415-1425, 10.5194/cp-7-1415-2011, 2011.

Ganopolski, A., and Roche, D. M.: On the nature of lead-lag relationships during glacial-interglacial climate transitions, Quat. Sci. Rev., 28, 3361-3378, 10.1016/j.quascirev.2009.09.019, 2009.

Heinze, C., Hoogakker, B. A. A., and Winguth, A.: Ocean carbon cycling during the past 130 000 years - a pilot study on inverse palaeoclimate record modelling, Clim. Past., 12, 1949-1978, 10.5194/cp-12-1949-2016, 2016.

Kobayashi, H., Abe-Ouchi, A., and Oka, A.: Role of Southern Ocean stratification in glacial atmospheric $CO_2$ reduction evaluated by a three-dimensional ocean general circulation model,

Paleoceanography, 30, 1202-1216, 10.1002/2015pa002786, 2015.

Lambert, F., Tagliabue, A., Shaffer, G., Lamy, F., Winckler, G., Farias, L., Gallardo, L., and De Pol-Holz, R.: Dust fluxes and iron fertilization in Holocene and Last Glacial Maximum climates, Geophys. Res. Lett., 42, 6014-6023, 10.1002/2015gl064250, 2015.

Mariotti, V., Paillard, D., Bopp, L., Roche, D. M., and Bouttes, N.: A coupled model for carbon and radiocarbon evolution during the last deglaciation, Geophys. Res. Lett., 43, 1306-1313, 10.1002/2015gl067489, 2016.

Muglia, J., and Schmittner, A.: Glacial Atlantic overturning increased by wind stress in climate models, Geophys. Res. Lett., 42, 9862-9869, 10.1002/2015gl064583, 2015.

Schmittner, A., and Somes, C. J.: Complementary constraints from carbon (C-13) and nitrogen (N-15) isotopes on the glacial ocean's soft-tissue biological pump, Paleoceanography, 31, 669-693, 10.1002/2015pa002905, 2016.

Skinner, L. C., Primeau, F., Freeman, E., de la Fuente, M., Goodwin, P. A., Gottschalk, J., Huang, E., McCave, I. N., Noble, T. L., and Scrivner, A. E.: Radiocarbon constraints on the glacial ocean circulation and its impact on atmospheric CO2, Nat. Commun., 8, 10, 10.1038/ncomms16010, 2017.

Weber, S. L., Drijfhout, S. S., Abe-Ouchi, A., Crucifix, M., Eby, M., Ganopolski, A., Murakami, S., Otto-Bliesner, B., and Peltier, W. R.: The modern and glacial overturning circulation in the Atlantic Ocean in PMIP coupled model simulations, Clim. Past., 3, 51-64, 2007.